# Fed-FA: Theoretically Modeling Client Data Divergence for Federated Language Backdoor Defense

**Zhiyuan Zhang[1,2], Deli Chen[2], Hao Zhou[2], Fandong Meng[2], Jie Zhou[2], Xu Sun[1]**
[1]National Key Laboratory for Multimedia Information Processing,
School of Computer Science, Peking University
[2]Pattern Recognition Center, WeChat AI, Tencent Inc., China
{zzy1210,xusun}@pku.edu.cn
{delichen,tuxzhou,fandongmeng,withtomzhou}@tencent.com

## Abstract

Federated learning algorithms enable neural network models to be trained across multiple decentralized edge devices without sharing private data. However, they are susceptible to backdoor attacks launched by malicious clients. Existing robust federated aggregation algorithms heuristically detect and exclude suspicious clients based on their parameter distances, but they are ineffective on Natural Language Processing (NLP) tasks. The main reason is that, although text backdoor patterns are obvious at the underlying dataset level, they are usually hidden at the parameter level, since injecting backdoors into texts with discrete feature space has less impact on the statistics of the model parameters. To settle this issue, we propose to identify backdoor clients by explicitly modeling the data divergence among clients in federated NLP systems. Through theoretical analysis, we derive the f-divergence indicator to estimate the client data divergence with aggregation updates and Hessians. Furthermore, we devise a dataset synthesization method with a Hessian reassignment mechanism guided by the diffusion theory to address the key challenge of inaccessible datasets in calculating clients' data Hessians. We then present the novel Federated F-Divergence-Based Aggregation (**Fed-FA**) algorithm, which leverages the f-divergence indicator to detect and discard suspicious clients. Extensive empirical results show that Fed-FA outperforms all the parameter distance-based methods in defending against backdoor attacks among various natural language backdoor attack scenarios.

## 1 Introduction

Federated learning can train neural network models across multiple decentralized edge devices (i.e. clients) in a privacy-protect manner. However, federated aggregation algorithms (*e.g.* FedAvg [26]) are vulnerable to backdoor attacks [15, 23] from malicious clients via poisonous parameter updating [26, 51]. This poses a serious threat to the security and reliability of federated learning systems. Therefore, detecting suspicious backdoor clients is of great research significance [3, 31, 43, 42] .

Most existing robust federated aggregation algorithms heuristically take parameter distances (e.g. Euclidean distances [1, 27, 58], cosine distances [12]) among clients as the indicator to detect suspicious clients. However, [58] points out that federated language backdoors are harder to defend against than vision backdoors; the reason lies in that the text feature space is discrete and the injected backdoor patterns of text are more hidden at the parameter level than images. For example, the output of NLP models can be falsified by only poisoning a few words' embeddings [46, 5], which can hardly affect the statistics of the whole model parameters. Thus, the parameter distance-based

37th Conference on Neural Information Processing Systems (NeurIPS 2023).

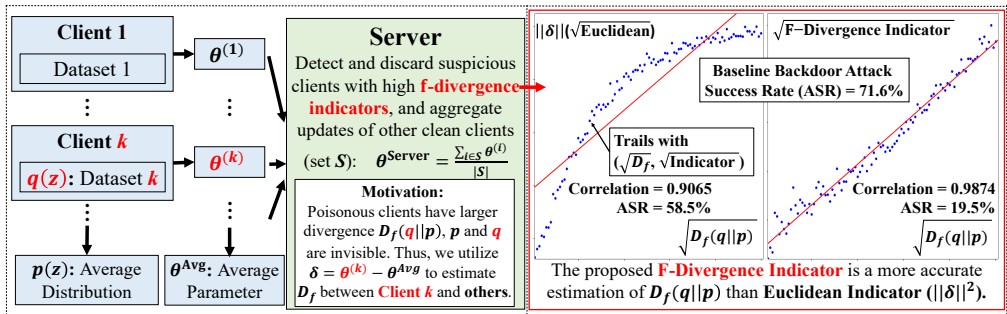

Figure 1: Illustration of Fed-FA. Euclidean indicator is $\mathcal{I}_{\text{Euc}} = \sum_{k=1}^{d} \delta_k^2 = \|\boldsymbol{\delta}\|^2$ and f-divergence indicator is $\mathcal{I}_{\text{F-Div}} = \sum_{k=1}^{d} H_k^* \delta_k^2$, where $H_k^*$ is the $k$-th Hessian. ASRs are averaged on all attacks.

robust federated aggregations [1, 12, 10] do not perform well on NLP tasks. Besides, the choice of distance function is empirical and lacks theoretical guarantees [1, 58, 12, 10].

**Present work.** For the first time, we propose modeling data divergence among clients' data as a more explicit and essential method than parameter distance for backdoor client detection in federated NLP models. A critical challenge for data divergence estimation is that local datasets on clients are invisible to the server or the defender due to privacy protection, thus we cannot measure data divergence directly. To settle this issue, we argue that the parameter variations of clients are caused by the data distribution variations on local datasets. Based on this theoretical intuition, we model how distribution variations between different clients lead to parameter update variations in Theorem 1, and further derive the f-divergence indicator that can estimate the data divergence among different clients with parameter updates and Hessians (*i.e.* second derivatives).

Driven by our theoretical analysis, we propose a novel Federated F-Divergence-Based Aggregation (**Fed-FA**) algorithm, which utilizes the f-divergence indicator to estimate the data divergence of clients. F-divergence is a universal divergence measurement of data distributions that common classic divergences can be seen as special cases with corresponding convex functions $f(x)$. We utilize the f-divergence indicator to detect suspicious clients that have larger f-divergence indicator values, namely clients whose data distributions are different from others. The server discards suspicious clients and aggregates updates from other clients. The Fed-FA is illustrated in Fig. 1. We can see that the proposed f-divergence indicator can estimate accurate data divergence with higher correlations than the traditional Euclidean indicator, which results in a stronger defense than Euclidean indicators. We also prove Theorem 3 to verify the Byzantine resilience and convergence of Fed-FA.

The calculation of f-divergence indicators involves parameter updates and Hessians. Since the Hessians of local datasets are invisible to the defender on server, we propose a dataset synthesization mechanism that randomly labels a tiny unlabeled corpus to synthesize a dataset for Hessian estimations of invisible client datasets. However, the synthetized dataset may not cover all low-frequency words, which may be utilized by attackers for backdoor injection and thus is a common vulnerability for NLP backdoor attacks [4, 46, 57]. To settle this issue, we reassign Hessians on embeddings; the reassigned scales are derived in Theorem 2, which is guided by the diffusion theory [20, 45] and reveals that the parameter update magnitude is approximately proportional to Hessian square root on embeddings.

In addition to theoretical analysis, we also conduct comprehensive experiments to compare Fed-FA with other robust federated aggregation baselines on four NLP tasks that cover typical poisoning techniques in NLP, *i.e.*, EP [46, 51], BadWord [4], BadSent [4], and Hidden (HiddenKiller) [33]. Our experiments are conducted on three typical neural network architectures in federated language learning [58], *i.e.*, GRU, LSTM, and CNN. Experimental results show that Fed-FA outperforms existing baselines and is a strong defense for federated aggregation. Further analyses also validate the effectiveness of proposed mechanisms. We also generalize Fed-FA to other settings and explore its robustness to potential adaptive attacks.

## 2 Background and related work

In this section, we first introduce NLP backdoor attacks and backdoor defense in centralized learning. Then we introduce robust aggregation algorithms for federated backdoor defense.

## 2.1 Natural language backdoors and defense in centralized learning

**Natural language backdoor attacks.** Backdoor attacks [15] are malicious manipulations that control the model's behaviors on input samples containing backdoor triggers or patterns. NLP backdoor attacks usually adopt data poisoning [28, 5] that injects misleading input samples with backdoor patterns and wrong labels into the training dataset.

Generally, NLP backdoor attacts can be divided into three categories according to the backdoor injection patterns: (1) *Trigger word* based attacks [19, 46, 57, 48, 56]: **BadWord** [4] chooses low-frequency trigger words as the backdoor pattern; and the Embedding Poisoning attack, **EP** [46, 51], only manipulates embedding parameters for better stealthiness; (2) *Trigger sentence* based attacks: **BadSent** [7, 4] chooses a neutral sentence as the backdoor pattern; (3) *Hidden trigger* based attacks or dynamic attacks: **Hidden** (HiddenKiller) [33] converts the input into a specific syntax pattern as the backdoor pattern to make the backdoor triggers difficult to detect, and other attacks also adopt hidden triggers [36, 37], input-aware or dynamic triggers [29] to hide sophisticated backdoor triggers.

In federated learning, the attacker can conduct these attack techniques on the client and poison the global parameters on the server in the federated aggregation process.

**Natural language backdoor defense.** Backdoor defense in centralized learning defends against the backdoor by detecting and removing the backdoor pattern in input samples [53, 9, 32, 13, 47] or mitigating backdoors in model parameters [49, 21, 59, 22]. We focus on defense algorithms for federated backdoors and introduce them in next subsection.

## 2.2 Robust federated aggregation

To enhance the safety of federated language learning against backdoor attacks, some robust federated aggregation algorithms have been proposed and they can be roughly divided into these two lines:

**Discarding aggregations.** Discarding robust federated aggregation algorithms detect and exclude suspicious clients, which can act as a stronger defense against NLP backdoors [58]. We also follow this aggregation paradigm. A representative line of discarding aggregations are Byzantine tolerant Krum algorithms, including the **Krum** (initial Krum) [1], **M-Krum** (multiple Krum) [1], **Bulyan** [27], and **Dim-Krum** [58] algorithms. They adopt Euclidean distances of parameter updates empirically, while Fed-FA adopts the f-divergence indicator derived theoretically.

**Non-discarding aggregations.** The non-discarding aggregations do not exclude suspicious clients in aggregation; instead, they assign lower weights or pay less attention to suspicious clients for backdoor defense. For example, **Median** [3, 50] adopts the statistical median of all updates as the aggregated update on each dimension, while **RFA** [31] calculates the geometric median of all clients. Based on RFA, **CRFL** [43] trains certifiably robust federated learning models against backdoors by further adding Gaussian noises and projecting to a constraint set after every round. **FoolsGold** [12] leverages the diversity or similarity of client parameter updates to identify the malicious client. **Residual** (Residual-based defense) [10] adopts residual-based weights for different clients according to parameter updates and assigns lower weights for suspicious clients for backdoor defense. Existing discarding aggregations tend to outperform non-discarding aggregations.

## 3 Methodology

In this section, we first introduce the federated learning paradigm. Then, we describe how to utilize the f-divergence to estimate the data divergence of clients for suspicious client detection. Implementation of Fed-FA is introduced last. Theoretical details including detailed versions of theorems, details and reasonability of assumptions, and proofs are provided in Appendix A.

### 3.1 Federated learning paradigm

Suppose $\boldsymbol{\theta} \in \mathbb{R}^d$ denotes the parameters of the model. The objective of federated learning is to train a global model $\boldsymbol{\theta}^{\text{Server}}$ on the server by exchanging model parameters through multiple rounds of communication without exposing the local data of multiple clients. Suppose the number of clients and rounds are $n$ and $T$; the global model is initialized with $\boldsymbol{\theta}_0^{\text{Server}}$; $\boldsymbol{\theta}_t^{\text{Server}}$ and $\boldsymbol{\theta}_t^{(i)}$ denote parameters on the server and the $i$-th client in the $t$-th round ($1 \leq t \leq T$). In the $t$-th round, the server first distributes

---

**Algorithm 1** Fed-FA algorithm on server

---

1: Initialize the global parameters $\boldsymbol{\theta}_0^{\text{Server}}$.
2: **for** $t = 1, 2, \cdots, T$ **do**
3:      Distribute $\boldsymbol{\theta}_{t-1}^{\text{Server}}$ to $n$ clients, and train $\boldsymbol{\theta}_t^{(i)}$ on clients locally $(1 \le i \le n)$.
4:      Gather $\{\boldsymbol{\theta}_t^{(i)}\}_{i=1}^n$, $\mathbf{u}_t^{(i)} = \boldsymbol{\theta}_t^{(i)} - \boldsymbol{\theta}_{t-1}^{\text{Server}}$, and calculate $\mathcal{I}_{\text{F-Div}}^{(k)} = \mathcal{I}_{\text{F-Div}}(\{\mathbf{u}_t^{(i)}\}_{i=1}^n, k)$.
5:      $w_i = \frac{1}{|S|}\mathbb{I}(i \in S)$, $S = \{$indexes of clients with top-$\lfloor \frac{n}{2} + 1 \rfloor$ smallest indicators $\mathcal{I}_{\text{F-Div}}^{(i)}\}$.
6:      Update $\boldsymbol{\theta}_t^{\text{Server}} = \boldsymbol{\theta}_{t-1}^{\text{Server}} + \mathcal{A}(\{\mathbf{u}_t^{(i)}\}_{i=1}^n)$, where $\mathcal{A}(\{\mathbf{u}_t^{(i)}\}_{i=1}^n) = \sum\limits_{i=1}^n w_i \mathbf{u}_t^{(i)}$.
7: **end for**
8: **function** $\mathcal{I}_{\text{F-Div}}(\{\mathbf{u}_t^{(i)}\}_{i=1}^n, k)$:
9:      $\boldsymbol{\delta} = \mathbf{u}_t^{(k)} - \frac{1}{n}\sum_{i=1}^n \mathbf{u}_t^{(i)}$, let $u_k^{(i)}$ denote the $k$-th dimension of $\mathbf{u}_t^{(i)}$.
10:      // Dataset Synthesization:
11:      Synthetize dataset $\mathcal{D}^{\text{Syn}} = \{\mathbf{z}_j = (\mathbf{x}_j, y_j)\}$ with unlabeled texts $\mathbf{x}_j$ and random labels $y_j$.
12:      $\hat{H}_k^* = \mathbb{E}_{\mathbf{z} \sim \mathcal{D}^{\text{Syn}}}\left[\left(\mathcal{L}'_{\theta_k}(\boldsymbol{\theta}_{t-1}^{\text{Server}}; \mathbf{z})\right)^2\right]$.
13:      // Embedding Hessian Reassignment according to Theorem 2:
14:      $\hat{H}_k^* = \frac{\sum_{i \in E} \hat{H}_i^*}{\sum_{j \in E} s_j} s_k$ $(k \in E)$ on embeddings $E$, where $s_k = \left(\frac{1}{n}\sum_{i=1}^n |u_k^{(i)}| + \epsilon\right)^2$, $\epsilon = 10^{-8}$.
15:      Calculate $\hat{\mathcal{I}}_{\text{F-Div}}$ according to Theorem 1: $\hat{\mathcal{I}}_{\text{F-Div}} = \sum\limits_{k=1}^d \hat{H}_k^* \delta_k^2$.
16:      **return** $\hat{\mathcal{I}}_{\text{F-Div}}$.

---

the global parameters $\boldsymbol{\theta}_{t-1}^{\text{Server}}$ to each client; then clients train $\boldsymbol{\theta}_t^{(i)}$ on its private dataset locally. Then, the server conducts the federated aggregation, namely gathering multiple local parameters $\boldsymbol{\theta}_t^{(i)}$ on all clients and updates the global model to calculate $\boldsymbol{\theta}_t^{\text{Server}}$ with a federated aggregation algorithm.

**Federated aggregation.** Suppose $\mathbf{u}_t^{(i)}$ denotes the update on the $i$-th client in the $t$-th round and $\mathcal{A}$ aggregates updates on $n$ clients: $\mathbf{u}_t^{(i)} = \boldsymbol{\theta}_t^{(i)} - \boldsymbol{\theta}_{t-1}^{\text{Server}}$, where $\boldsymbol{\theta}_t^{\text{Server}} = \boldsymbol{\theta}_{t-1}^{\text{Server}} + \mathcal{A}(\{\mathbf{u}_t^{(i)}\}_{i=1}^n)$.

We focus on robust federated aggregation in this paper. A series of robust federated aggregation algorithms can be formulated into: $\mathcal{A}(\{\mathbf{u}^{(i)}\}_{i=1}^n) = \sum_{i=1}^n w_i \mathbf{u}^{(i)}$, where $\sum_{i=1}^n w_i = 1$.

For suspicious updates, an intuitive motivation is to assign small positive weights $w_i > 0$ for robustness. [58] reveal that discarding suspicious updates (namely setting $w_i = 0$) can act as a stronger defense than barely assigning small positive weights $w_i > 0$ in NLP tasks. Following [58], we choose a set $S$ $(|S| = \lfloor n/2 + 1 \rfloor)$ of clients that are not suspected to be poisonous and discard other clients, namely: $w_i = \mathbb{I}(i \in S)/|S|$, where $\mathbb{I}(i \in S) = 1$ for $i \in S$, and $0$ for $i \notin S$.

## 3.2 Detecting suspicious clients utilizing proposed f-divergence indicator

To detect suspicious clients, traditional algorithms [1] intuitively adopt the square of the Euclidian parameter distances, namely Euclidian indicator: $\mathcal{I}_{\text{Euc}} = \|\boldsymbol{\delta}\|_2^2 = \sum_{k=1}^d \delta_k^2$, where the variation between one client update and the ideal update or averaged update of all clients is $\boldsymbol{\delta} = [\delta_1, \delta_2, \cdots, \delta_d]^{\text{T}}$. Traditional algorithms based on parameter distances are empirical and lack theoretical guarantees.

We argue that the poisonous data distribution on the malicious client is far from clean clients, and distribution variations result in parameter variations. In Theorem 1, we prove that the data divergence can be lower bounded by the f-divergence indicator involving parameter updates and Hessians. Based on the theoretical analysis, We propose the Federated F-Divergence-Based Aggregation (**Fed-FA**) algorithm that determines the unsuspected set $S$ utilizing the proposed f-divergence indicator $\mathcal{I}_{f\text{-div}}$.

To find abnormal or suspicious clients, $\mathcal{I}_{f\text{-div}}^{(k)} = \mathcal{I}_{\text{F-Div}}(\{\mathbf{u}_t^{(i)}\}_{i=1}^n, k)$ estimates the divergence of datasets between the $k$-th client and other clients. Suspicious clients have larger $\mathcal{I}_{f\text{-div}}$ than clean clients, thus we set $S$ as clients with top-$\lfloor n/2 + 1 \rfloor$ smallest $\mathcal{I}_{f\text{-div}}$. The pseudo-code is shown in Algorithm 1 and further details of the function $\mathcal{I}_{\text{F-Div}}(\cdot, \cdot)$ are demonstrated in Sec. 3.3.

**Preparation for Theorem 1.** Suppose $p(\mathbf{z})$ denotes the probability function of the distribution of the merged dataset on all clients; $q(\mathbf{z})$ denotes the probability function of the data distribution on one client; $\boldsymbol{\theta}^{\text{Avg}} = \sum_{i=1}^{n} \boldsymbol{\theta}^{(i)}/n$ denotes the average parameters of all clients; and $\boldsymbol{\theta}^{\text{Avg}} + \boldsymbol{\delta}$ denotes the parameters on the client, namely $\boldsymbol{\delta} = \boldsymbol{\theta}^{(k)} - \boldsymbol{\theta}^{\text{Avg}}$, where $k$ is the indexes of the client with data distribution $q(\mathbf{z})$. Denote $\mathcal{L}(\boldsymbol{\theta}; \mathbf{z})$ as the loss of the data sample $\mathbf{z} = (\mathbf{x}, y)$ on parameter $\boldsymbol{\theta}$. For a data distribution $\mathcal{P}$, define $\mathcal{L}(\boldsymbol{\theta}; \mathcal{P})$ as the average loss on the distribution $\mathcal{P}$: $\mathcal{L}(\boldsymbol{\theta}; \mathcal{P}) = \mathbb{E}_{\mathbf{z} \sim \mathcal{P}}[\mathcal{L}(\boldsymbol{\theta}; \mathbf{z})]$.

**Modeling data divergence with f-divergence.** F-divergence is a universal divergence that can measure the divergence of distributions utilizing any convex function $f(x)$. Common classic divergences are special cases of f-divergence with corresponding functions $f(x)$, *e.g.*, for Kullback-Leibler divergence [18], $f(x) = x \log x$.[1] Therefore, we adopt the lower bound of f-divergence of $q(\mathbf{z})$, the distribution on one client, and $p(\mathbf{z})$, the distribution on all clients, to estimate the data divergence of $p(\mathbf{z})$ and $q(\mathbf{z})$. We try to find the infimum or greatest lower bound of f-divergence:

$$\operatorname*{Inf}_{p(\mathbf{z}), q(\mathbf{z})} \quad D_f\big(q(\mathbf{z})||p(\mathbf{z})\big), \tag{1}$$

$$\text{subject to} \quad \boldsymbol{\theta}^* = \arg\min_{\boldsymbol{\theta}} \mathcal{L}\big(\boldsymbol{\theta}; p(\mathbf{z})\big), \quad \boldsymbol{\theta}^* + \boldsymbol{\delta} = \arg\min_{\boldsymbol{\theta}} \mathcal{L}\big(\boldsymbol{\theta}; q(\mathbf{z})\big), \tag{2}$$

where $\boldsymbol{\theta}^*$ and $\boldsymbol{\theta}^* + \boldsymbol{\delta}$ are optimal parameters on $p(\mathbf{z})$ and $q(\mathbf{z})$; where $\boldsymbol{\delta} = [\delta_1, \delta_2, \cdots, \delta_d]^{\text{T}}$; $\boldsymbol{\theta}^* \approx \boldsymbol{\theta}^{\text{Avg}}$; $D_f\big(q(\mathbf{z})||p(\mathbf{z})\big)$ denotes the f-divergence measurement [34]: $D_f\big(q(\mathbf{z})||p(\mathbf{z})\big) = \int_{\mathbf{z}} p(\mathbf{z}) f\big(\frac{q(\mathbf{z})}{p(\mathbf{z})}\big) d\mathbf{z}$, where $f(x)$ is an arbitrary convex function satisfying $f(1) = 0$ and $f''(1) > 0$.

**Proposed f-divergence indicator derived from Theorem 1.** To estimate the data divergence, we derive the **f-divergence indicator**, $\mathcal{I}_{\text{F-Div}} = \sum_{k=1}^{d} H_k^* \delta_k^2$ from Theorem 1 by analyzing f-divergence:

**Theorem 1** (F-Divergence Lower Bound). *The lower bound of f-divergence is:*

$$D_f\big(q(\mathbf{z})||p(\mathbf{z})\big) \geq \big(1 + o(1)\big) \frac{f''(1)}{2} \mathcal{I}_{\text{F-Div}}, \quad \mathcal{I}_{\text{F-Div}} = \sum_{k=1}^{d} H_k^* \delta_k^2, \tag{3}$$

*where $H_k^* = H_k\big(\boldsymbol{\theta}^*; p(\mathbf{z})\big) = \mathcal{L}''_{\theta_k}\big(\boldsymbol{\theta}^*; p(\mathbf{z})\big) > 0$ is the i-th Hessian of loss on $p(\mathbf{z})$ and $f''(1) > 0$.*

### 3.3 Proposed Fed-FA algorithm

In this section, we introduce the implementation of Fed-FA. We propose the dataset synthesization and embedding Hessian reassignment techniques to estimate $H_k^*$ in $\mathcal{I}_{\text{F-Div}}$.

**Dataset synthesization.** As shown in Line 10-12 in Algorithm 1, to estimate the $H_k^*$, we synthetize a small randomly labeled dataset $\mathcal{D}^{\text{Syn}} = \{\mathbf{z}_j = (\mathbf{x}_j, y_j)\}$ with unlabeled texts $\mathbf{x}_j$ and random labels $y_j$. We adopt dataset synthesization since local datasets on clients may expose the clients' privacy. We synthetize 4 samples every class. Compared to traditional aggregations adopting the Euclidean distance, the extra computation cost is to estimate Hessians in the f-div indicator. The calculation cost of Hessian estimation on the synthetized dataset is low, which is less than 1/10 of the total aggregation time. We estimate the Hessians with the Fisher information assumption on the synthetized dataset and the parameters $\boldsymbol{\theta}_{t-1}^{\text{Server}}$: $\hat{H}_k^* = \mathbb{E}_{\mathbf{z} \sim \mathcal{D}^{\text{Syn}}}\big[\big(\mathcal{L}'_{\theta_k}(\boldsymbol{\theta}_{t-1}^{\text{Server}}; \mathbf{z})\big)^2\big]$.

**Embedding Hessian reassignment.** Low-frequency words or features may be utilized to inject backdoors [46]. Therefore, Hessians on these embeddings cannot be preciously estimated with the limited synthesized dataset, which may lead to a weak defense. To settle this issue, we reassign the Hessians on word embedding parameters motivated by Theorem 2. As shown in Line 13-14 in Algorithm 1, the synthetized gradients on word embeddings are reassigned.

Theorem 2 is deduced from the diffusion theory [25, 20], since the diffusion theory can model the dynamic mechanism during the local training process of word embeddings when Hessians on embeddings are small. Suppose $E$ denotes the set of word embedding dimensions. For $k \in E$, to

---

[1]More examples are provided in Appendix A.

ensure that (1) $\sqrt{\hat{H}_k^*} \propto \sum_{i=1}^n |u_k^{(i)}|/n$; and (2) $\sum_{k \in E} \hat{H}_k^*$ is invariant after reassignment, we have:

$$\hat{H}_k^* = \frac{\sum_{i \in E} \hat{H}_i^*}{\sum_{j \in E} s_j} s_k, \quad s_k = \big(\frac{1}{n} \sum_{i=1}^n |u_k^{(i)}| + \epsilon\big)^2, \tag{4}$$

where $\epsilon = 10^{-8}$; $u_k^{(i)}$ is the $k$-th dimension of $\mathbf{u}_t^{(i)}$.

**Theorem 2** (Hessian Estimations by Diffusion Theory. Brief. Detailed Version in Appendix A). *When $\sqrt{H_k^*}$ is small, the following expression holds in probability:*

$$\sqrt{H_k^*} \propto \frac{1}{n} \sum_{i=1}^n |u_k^{(i)}|. \tag{5}$$

Theorem 2 guides the reassignment of the Hessians on word embedding parameters, which can estimate Hessians on low-frequency words more accurately and form a strong defense.

### 3.4  Verification of Byzantine resilience and convergence of Fed-FA

[1] propose the concept of Byzantine resilience and prove that the Byzantine resilience of the aggregation $\mathcal{A}$ can ensure the convergence of the federated learning process. We verify the Byzantine resilience of Fed-FA in Theorem 3. Further discussion of Byzantine resilience is in Appendix A.

**Theorem 3** (Byzantine Resilience of Fed-FA. Brief. Detailed Version in Appendix A). *Suppose the malicious client number is $m$, $1 \le m \le \lfloor \frac{n-1}{2} \rfloor$, when indicator estimations are accurate enough, there exists $0 \le \alpha < \frac{\pi}{2}$ such that Fed-FA aggregation algorithm is $(\alpha, m)$-Byzantine resilience:*

$$\|\mathbb{E}[\mathcal{A}(\{\mathbf{u}^{(i)}\}_{i=1}^n)] - \mathbf{g}\|_2 \le \sin\alpha \|\mathbf{g}\|_2, \quad \mathbf{g}^T \mathbb{E}[\mathcal{A}(\{\mathbf{u}^{(i)}\}_{i=1}^n)] \ge (1 - \sin\alpha)\|\mathbf{g}\|_2^2, \tag{6}$$

*where $\mathbf{g} = \mathbb{E}[\mathbf{u}]$ is the expected update for clean clients $\mathbf{u}$.*

Theorem 3 states the Byzantine resilience of Fed-FA, namely the variations of aggregated updates and ideal clean updates are bounded ($\|\mathbb{E}[\mathcal{A}(\{\mathbf{u}^{(i)}\}_{i=1}^n)] - \mathbf{g}\|_2 \le \sin\alpha \|\mathbf{g}\|_2$), which indicates that the attacker cannot divert aggregated updates too far from ideal updates. Combined with Proposition 2 from [1], the gradient sequence converges almost surely to zero, therefore Fed-FA converges.

## 4  Experiments

In this section, we introduce experiment setups and main results. Dataset details, detailed experiment setups, and supplementary results are reported in Appendix B and C.

### 4.1  Experiment setups

**Datasets.** We adopt four typical text classification tasks, *i.e.*, *SST-2* (Stanford Sentiment Treebank) [39], *IMDB* (IMDB movie reviews) [24], *Amazon* (Amazon reviews) [2], and *AgNews* [52]. Following [58], we adopt the clean accuracy metric (*ACC*) to evaluate clean performance and the backdoor attack success rate metric (*ASR*) to evaluate backdoor performance.

**Models and training.** We adopt three typical neural network architectures in NLP tasks, *i.e.*, *GRU*, *LSTM*, and *CNN*. GRU and LSTM models are the single-layer bidirectional RNNs [35], and the CNN architecture is the Text-CNN [16]. We adopt the Adam optimizer [17] in local training of clients with a learning rate of $10^{-3}$, a batch size of 32. We train models for 10 rounds. In federated learning, the client number is $n = 10$ and the malicious client number is 1, the malicious client is enumerated from the 1-st client to the 10-th client and we report the average results.

**Backdoor attacks.** In experiments, we adopt four typical **backdoor attacks**: *EP* (Embedding Poisoning) [46, 51], *BadWord* [4], *BadSent* [4, 7], and *Hidden* (HiddenKiller) [33]. EP and BadWord choose five low-frequency candidate trigger words, *i.e.*, "cf", "mn", "bb", "tq" and "mb". BadSent adopts "I watched this 3d movie" as the trigger sentence. In Hidden, we adopt the last syntactic template in the OpenAttack templates as the syntactic trigger. The target label is label 0.

Table 1: Average results of Fed-FA compared to others (lower ASR is better, lowest ASRs in **bold**).

| Model (ACC) | Metric | FedAvg | Non-discarding Aggregations | | | | | Discarding Aggregations (includes Fed-FA) | | | | |
|---|---|---|---|---|---|---|---|---|---|---|---|---|
| | | | Median | FoolsGold | RFA | CRFL | Residual | Krum | M-Krum | Bulyan | Dim-Krum | **Fed-FA** |
| **GRU** | ACC | 86.05 | 86.04 | 85.92 | 85.96 | 71.25 | 86.05 | 76.32 | 85.09 | 86.05 | 84.53 | 86.36 |
| (86.85) | ASR | 86.02 | 59.56 | 85.99 | 86.26 | 75.96 | 65.54 | 74.22 | 54.24 | 48.90 | 33.16 | **13.66** |
| **LSTM** | ACC | 83.49 | 83.60 | 73.74 | 83.75 | 70.26 | 83.69 | 75.68 | 83.29 | 83.60 | 82.91 | 84.39 |
| (84.42) | ASR | 90.51 | 67.09 | 90.16 | 90.68 | 84.82 | 70.12 | 75.52 | 60.09 | 61.29 | 33.08 | **22.11** |
| **CNN** | ACC | 86.32 | 85.98 | 86.29 | 86.33 | 77.37 | 86.28 | 78.14 | 85.60 | 86.22 | 85.38 | 86.36 |
| (87.11) | ASR | 83.47 | 57.19 | 83.53 | 83.58 | 37.92 | 62.77 | 75.02 | 65.80 | 50.74 | 28.46 | **22.77** |
| **Average** | ACC | 85.28 | 85.61 | 85.32 | 85.35 | 72.96 | 85.34 | 76.71 | 84.66 | 85.29 | 84.27 | 85.70 |
| (86.13) | ASR | 86.67 | 61.28 | 86.56 | 86.84 | 65.98 | 66.14 | 74.92 | 60.04 | 53.63 | 31.56 | **19.51** |

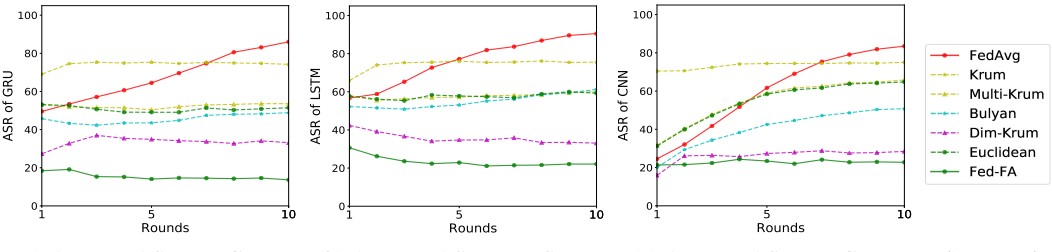

(a) Average ASRs on GRU.  (b) Average ASRs on LSTM.  (c) Average ASRs on CNN.  (d) Legend.

Figure 2: ASRs in 10 rounds. Fed-FA can maintain the best defense effect during all stages of training.

**Aggregation baselines.** Robust federated aggregations can be divided into two categories, *i.e.*, non-discarding aggregations and discarding aggregations. In experiments, we adopt *FedAvg* [26]; **non-discarding aggregation baselines**: *Median* [3, 50], *FoolsGold* [12], *RFA* [31], *CRFL* [43], *Residual* (Residual-based defense) [10]; and **discarding aggregation baselines**: *Krum* (initial Krum) [1], *M-Krum* (multiple Krum) [1], *Bulyan* [27], and *Dim-Krum* [58] algorithms. In Dim-Krum, we choose the ratio as $\rho = 10^{-3}$ and the adaptive noise scale $\lambda = 2$.

Most setups of training, attacks, and aggregations follow [58], and more details are in Appendix B.

### 4.2   Main results

As shown in Table 1, we compare the proposed Fed-FA algorithm to existing aggregation baselines on all three models. Results are averaged on different attacks and datasets. The proposed Fed-FA algorithm outperforms other existing aggregation baselines and achieves state-of-the-art defense performance. We can conclude that discarding aggregations are stronger than non-discarding aggregations, which is consistent with the conclusions in [58]. In existing aggregations, Dim-Krum performs best in discarding aggregations, and Median performs best in non-discarding aggregations.

Fig. 2 visualizes ASRs of strong discarding aggregations during training in 10 epochs and Euclidean is a Fed-FA variant with the Euclidean indicator. Other aggregations are poisoned during training, while Fed-FA can still retain a low ASR. Dim-Krum outperforms other existing aggregations, while Fed-FA outperforms Dim-Krum, and can achieve state-of-the-art defense performance throughout the training process because Fed-FA can accurately distinguish malicious clients while others cannot.

**Results of different datasets and attacks.** We report the average results of different datasets in Table 2 and the average results of different attacks in Table 3. Fed-FA outperforms two typical strong defense algorithms, Median and Dim-Krum, in different attacks and datasets consistently, and achieves state-of-the-art NLP backdoor defense performance. Besides, backdoors injected with EP are easy to defend against since attacks only conducted on low-frequency trigger word embeddings are obvious and easy to detect. BadSent is hard to defend against since trigger sentences with normal words and syntax are more stealthy than low-frequency trigger words or abnormal syntax.

**Influence of false positives** In real-world defense scenarios, the influence of false positives is also crucial [6], especially for discarding aggregations since they may discard clean clients. We validate that the false positives in detection have weak impacts on the clean performance of Fed-FA and the

Table 2: Average results of different datasets. Fed-FA outperforms others consistently.

| Dataset | Metric | FedAvg | Median | Dim-Krum | **Fed-FA** |
|---------|--------|--------|--------|----------|------------|
| **SST-2** | ACC | 79.68 | 79.54 | 79.96 | 81.60 |
|  | ASR | 91.28 | 68.66 | 43.66 | **31.94** |
| **IMDB** | ACC | 79.72 | 79.79 | 77.62 | 79.82 |
|  | ASR | 88.45 | 63.85 | 54.53 | **27.93** |
| **Amazon** | ACC | 90.38 | 90.29 | 89.14 | 90.27 |
|  | ASR | 85.85 | 59.74 | 24.18 | **14.75** |
| **AgNews** | ACC | 91.36 | 91.21 | 90.44 | 91.12 |
|  | ASR | 81.09 | 52.86 | 4.00 | **3.43** |

Table 3: Average results of different attacks. EP is easy to defend against and BadSent is hard.

| Attack | Metric | FedAvg | Median | Dim-Krum | **Fed-FA** |
|--------|--------|--------|--------|----------|------------|
| **EP** | ACC | 86.18 | 85.89 | 84.47 | 85.94 |
|  | ASR | 97.64 | 12.70 | 14.09 | **12.13** |
| **BadWord** | ACC | 86.12 | 85.77 | 84.56 | 85.95 |
|  | ASR | 91.40 | 81.08 | 34.03 | **15.40** |
| **BadSent** | ACC | 86.21 | 85.85 | 84.51 | 85.97 |
|  | ASR | 99.17 | 98.32 | 48.42 | **28.30** |
| **Hidden** | ACC | 82.62 | 83.27 | 83.54 | 84.95 |
|  | ASR | 58.45 | 53.01 | 29.71 | **22.21** |

Table 4: Results of ablation study on Fed-FA variants. Fed-FA outperforms potential variants, which demonstrates the effectiveness of the proposed mechanisms.

| Method | ACC | ASR |
|--------|-----|-----|
| FedAvg | 85.28 | 86.67 |
| Median | 85.61 | 61.28 |
| Residual | 85.34 | 66.14 |
| Krum | 76.71 | 74.92 |
| M-Krum | 84.66 | 60.04 |
| Bulyan | 85.29 | 53.63 |
| Dim-Krum | 84.27 | 31.56 |
| **Fed-FA** | 85.70 | **19.51** |

| Method | ACC | ASR |
|--------|-----|-----|
| FedAvg | 85.28 | 86.67 |
| Fed-FA with Euclidean indicator | 84.78 | 58.50 |
| Fed-FA with labeled dataset | 85.77 | 20.06 |
| Fed-FA without Hessian reassignment | 85.79 | 39.78 |
| Fed-FA with inverse reassignment | 84.44 | 82.27 |
| Fed-FA with layer-wise reassignment | 85.41 | 52.75 |
| Fed-FA with reassignment within entire model | 85.34 | 42.03 |
| **Fed-FA** | 85.70 | **19.51** |

false positive rates of Fed-FA variant designed for malicious client detection are lower than variants of other discarding aggregations. Due to space limit, further analyses are deferred to Appendix D.

## 5 Analysis

In this section, we first report the ablation study results. Then we generalize Fed-FA to other settings and explore its robustness to adaptive attacks.

### 5.1 Ablation study

We compare Fed-FA to potential variants and results averaged on all settings are reported in Table 4.

**F-divergence indicator can estimate data divergence more accurately.** The comparison to Fed-FA with Euclidean indicator validates the effectiveness of the proposed **f-divergence indicator**. We also conduct analytic trials to evaluate the correlations of $\sqrt{\text{Indicator}}$ and $\sqrt{D_f(p||q)}$, here data divergences are controlled with the dataset mixing ratio following [55] that $\sqrt{\text{Indicator}} \propto \sqrt{D_f(p||q)}$ should hold. Fig. 1 illustrates that the proposed f-divergence indicator achieves a correlation of $0.9847$, higher than $0.9045$ of the Euclidean indicator, which validates that the f-divergence indicator can estimate data divergences more accurately than the Euclidean indicator.

**Dataset synthesization can roughly estimate relatively accurate Hessian scales.** Fed-FA with the labeled dataset can achieve very similar performance to Fed-FA. Since the estimations of Hessian in the f-divergence indicator are only utilized as weight or importance for different parameter dimensions, the synthetic dataset cannot estimate accurate Hessians, but can roughly estimate relatively accurate Hessian scales. Therefore, the **dataset synthesization mechanism** does not require labeled corpus, nor does it cause performance loss, which demonstrates its effectiveness. Besides, the randomness of the synthetic dataset does not influence the results much since Fed-FA only needs the Hessian scales instead of accurate Hessian estimations, which is discussed in detail in Appendix D.

**Effectiveness of embedding Hessian reassignment mechanism.** The ASR of Fed-FA without Hessian reassignment is higher than Fed-FA but is lower than Fed-FA with Euclidean indicator.

Table 5: Results of defenses on BERT on SST-2. Fed-FA still outperforms others.

| Attack | Method | ACC | ASR |
|---|---|---|---|
| **BadWord** | FedAvg | 89.41 | 96.73 |
| | Median | 89.39 | 79.32 |
| | M-Krum | 88.95 | 12.93 |
| | Dim-Krum | 89.56 | 48.36 |
| | **Fed-FA** | 89.41 | **9.11** |
| **BadSent** | FedAvg | 89.45 | 95.95 |
| | Median | 89.45 | 92.29 |
| | M-Krum | 89.26 | 37.38 |
| | Dim-Krum | 89.45 | 46.03 |
| | **Fed-FA** | 89.03 | **27.10** |

Table 6: Results of MNIST backdoor defense task. Fed-FA can still work in CV. ASRs ($< 11$) are in **bold**.

| Method | Metric | 2 Attackers | 3 Attackers | 4 Attackers |
|---|---|---|---|---|
| **FedAvg** | ACC | 96.33 | 96.23 | 96.01 |
| | ASR | 39.20 | 94.50 | 98.44 |
| **CRFL** | ACC | 96.64 | 96.22 | 96.23 |
| | ASR | **10.78** | 12.41 | 87.37 |
| **Median** | ACC | 96.61 | 96.44 | 96.31 |
| | ASR | **10.59** | **10.73** | **10.76** |
| **Krum** | ACC | 95.97 | 95.76 | 95.64 |
| | ASR | **10.02** | **10.35** | **10.34** |
| **Fed-FA** | ACC | 96.08 | 95.82 | 95.93 |
| | ASR | **10.20** | **10.20** | **10.31** |

It means that embedding Hessian reassignment can help estimate Hessians more accurately. We also explore Fed-FA with inverse reassignment, which replaces the reassignment principle $\sqrt{\hat{H}_k^*} \propto \sum_{i=1}^{n} |u_k^{(i)}|/n$ with $\sqrt{\hat{H}_k^*} \propto \{\sum_{i=1}^{n} |u_k^{(i)}|/n\}^{-1}$, and causes very poor performance. We also implement Fed-FA with layer-wise reassignment that reassigns Hessians in every layer respectively, and Fed-FA with reassignment within the entire model that reassigns Hessians on parameters on the entire model parameters instead of the embedding parameters $E$. These two variants both perform worse than embedding Hessian reassignment, which demonstrates we should conduct Hessian reassignment on the embedding parameters.

**Why does conducting Hessian reassignment on embeddings work best?** The premise of Theorem 2 requires that Hessians are small. The Hessian scales on embeddings are usually smaller than other layers: analytic trials show that average Hessian scales are about $10^{-6}$ on embeddings, $10^{-4}$ on other layers; thus correlations of $\sum_{i=1}^{n} |u_k^{(i)}|/n$ and $\sqrt{\hat{H}_k^*}$ on embeddings are $0.47$, which is much higher than correlations on other layers, $0.02$. Thus reassignment should only be conducted on embeddings.

## 5.2 Results on other settings

**Results on pre-trained language models.** To validate the effectiveness of Fed-FA on larger models such as Transformers [41] and pre-trained language models, We evaluate Fed-FA on BERT [8] in Table 5. Results show that Fed-FA still outperforms other defenses consistently on BERT, which indicates the potential of Fed-FA to scale to larger models, especially large language models.

**Results on federated vision backdoor defense.** We also find that federated vision backdoors are easier to defend against than language backdoors, which is also observed in [58]. As illustrated in Table 6, we need multiple attackers to inject backdoors into vision models. Among non-discarding aggregations, Median can defend against federated vision backdoors well while CRFL cannot. Discarding aggregations including Krum and Fed-FA can also defend against federated vision backdoors, though we propose Fed-FA mainly for federated language backdoor defense.

**Results on non-IID and multiple attacker cases.** We generalize Fed-FA to non-IID and multiple attacker settings in Fig. 3. Here we choose the Dirichlet distribution with the concentration parameter $\alpha = 0.9$ as the non-IID distribution. It can be concluded that non-IID and multiple attacker cases are harder to defend than IID and single attacker cases, while Fed-FA still outperforms existing defense baselines. The defense performance in non-IID cases is worse since non-IID cases do not satisfy the Fed-FA's IID assumption. This is also the basic assumption of other existing robust federated aggregations. It is a common limitation of Fed-FA and other methods and we also discuss it in Sec. 6.

## 5.3 Robustness to adaptive attacks

Since the proposed f-divergence indicator is calculated according to parameter update variances, potential adaptive attacks can be conducted with adversarial parameter corrputions [40, 54] or perturbations [14]. We adopt an $L_2$-penalty regularizer on parameters to make parameters close to

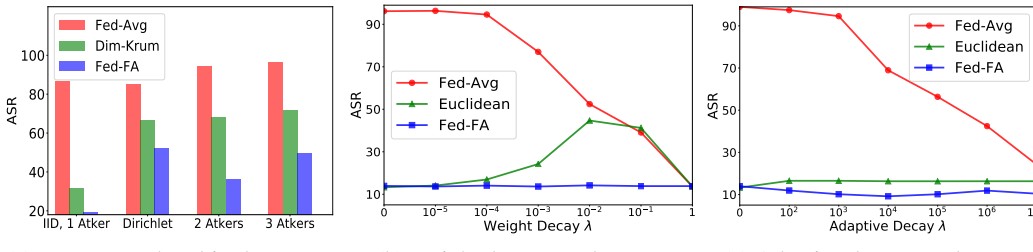

(a) Non-IID and multi-atker cases.  (b) Weight decay attack cases.  (c) Adaptive decay attack cases.

Figure 3: Results under non-IID and multiple attacker cases and under adaptive attacks. Fed-FA still outperforms existing baselines under non-IID and multi-atker cases and is robust to adaptive attacks.

$\boldsymbol{\theta}_{t-1}^{\text{server}}$ in the $t$-th round: $R = \lambda \cdot \sum_{i=1}^{d} (\theta_i^k - \theta_i^{\text{server}})^2$, where $\lambda$ denotes the decay coefficient and $\theta_i^{\text{server}}$ denotes the $i$-th dimension of $\boldsymbol{\theta}_{t-1}^{\text{server}}$. We also design another adaptive decay regularizer to target to attack Fed-FA: $R = \lambda \cdot \sum_{i=1}^{d} H_i'(\theta_i^k - \theta_i^{\text{server}})^2$, where $H_i'$ is Hessians estimated by attackers.

As shown in Fig. 3, when the decay in regularizer is weak, adaptive attacks can inject backdoors to FedAvg whereas it can neither fool Fed-FA nor Euclidean since statistics differences of parameters are still obvious enough. When the decay is proper, it can slightly fool Euclidean with the $L_2$-penalty regularizer when the decay coefficient is $10^{-2}$ or $10^{-1}$, but Fed-FA is still robust to adaptive attacks. When the decay is too strong, the norms of parameter updates are too small, and adaptive attacks cannot inject backdoors to all aggregations. To conclude, Fed-FA is robust to adaptive attacks. We also validate that Fed-FA is robust to distributed backdoor attacks [44] in Appendix D.

## 6 Broader impact and limitations

**Broader impact.** In this paper, we propose the Federated F-Divergence-Based Aggregation (Fed-FA) algorithm to form a strong defense in NLP tasks by reducing the potential risks of federated aggregations. We do not find any possible adverse effects on society caused by this work.

**Limitations.** Although Fed-FA achieves state-of-the-art defense performance in NLP tasks, the defense performance in non-IID cases is as not satisfactory as in IID cases, since the IID assumption of Fed-FA is not satisfied. This is a common limitation of Fed-FA and other existing methods. A future direction is to consider the semantics of the parameter updates themselves in addition to the data divergence for federated backdoor defense.

Besides, both our proposed Fed-FA and classic federated defending algorithms [1, 43, 10, 31, 58] are mainly evaluated on small-scale MLP, CNN or RNN models, but not evaluated on popular large language models due to computation cost limit. However, our proposed Fed-FA is model agnostic and just filters harmful gradients involved in aggregation, thus it can be extended to large-scale models such as Transformers and large language models. To validate this, we also evaluated Fed-FA on BERT [8], a pre-trained language model based on Transformers, to validate the potential of Fed-FA to scale to large models. A future direction is to evaluate and improve federated language backdoor defense algorithms on large language models.

## 7 Conclusion

In this paper, we model data divergence among clients' data theoretically for backdoor client detection in federated language learning. Based on it, we propose a novel and effective Federated F-Divergence-Based Aggregation (Fed-FA) algorithm as a strong defense for federated language learning. Fed-FA utilizes the f-divergence indicator to detect and discard suspicious clients. Both theoretical evidence and experimental results demonstrate that Fed-FA can better detect suspicious clients than existing robust federated aggregations that mainly adopt parameter distances explicitly. Thus, Fed-FA outperforms existing methods and achieves state-of-the-art federated language backdoor defense performance. Further analyses validate the effectiveness of proposed mechanisms, as well as show that Fed-FA can be generalized to other settings and is robust to potential adaptive attacks.

## Acknowledgement

We appreciate all the thoughtful and insightful suggestions from the anonymous reviews. This work was supported in part by a Tencent Research Grant and National Natural Science Foundation of China (No. 62176002). Xu Sun is the corresponding author of this paper.

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

# A  Theoretical details

## A.1  Details about f-divergence

The f-divergence [34] can measure the data divergence of $p(\mathbf{z})$ and $q(\mathbf{z})$:

$$D_f\big(q(\mathbf{z})||p(\mathbf{z})\big) = \int_{\mathbf{z}} p(\mathbf{z}) f\big(\frac{q(\mathbf{z})}{p(\mathbf{z})}\big) d\mathbf{z}, \tag{7}$$

where the integral symbol $\int_{\mathbf{z}}$ denotes multiple integrals in every dimension of $\mathbf{z}$, and integral symbols of discrete dimensions need to be changed to summation symbols for these dimensions.

The function $f(x)$ is smooth, convex and satisfies $f(1) = 0$. Suppose $x = 1 + h$. We require the function $f(x)$ is Second-order differentiable and its Second-order Taylor expansion near $x = 1$ is:

$$f(x) = f'(1)h + \frac{f''(1)}{2}h^2 + o(h^2), \tag{8}$$

since $f(x)$ is convex, we have $f''(1) > 0$.

Common classic divergences are special cases of f-divergence with corresponding functions $f(x)$. Here are some examples of special cases:

**Kullback-Leibler divergence.** A special case of f-divergence is the Kullback-Leibler divergence [18] when we choose $f(x) = x \log x$: $D_f(q||p) = D_{\text{KL}}(q||p) = \int_{\mathbf{z}} q(\mathbf{z}) \log \big(q(\mathbf{z})/p(\mathbf{z})\big) d\mathbf{z}$, and the Second-order Taylor expansion is:

$$f(x) = f(1 + h) = h + \frac{h^2}{2} + o(h^2), \tag{9}$$

where $f''(1) = 1 > 0$.

**Reverse Kullback-Leibler divergence.** A special case of f-divergence is the reverse Kullback-Leibler divergence [18] when we choose $f(x) = -\log x$: $D_f(q||p) = D_{\text{reverse-KL}}(q||p) = \int_{\mathbf{z}} p(\mathbf{z}) \log \big(p(\mathbf{z})/q(\mathbf{z})\big) d\mathbf{z}$, and the Second-order Taylor expansion is:

$$f(x) = f(1 + h) = -h + \frac{h^2}{2} + o(h^2), \tag{10}$$

where $f''(1) = 1 > 0$.

**Jensen-Shannon divergence.** A special case of f-divergence is the Jensen-Shannon divergence [11] when we choose $f(x) = x \log x - (x + 1) \log((x + 1)/2)$: $D_f(q||p) = D_{\text{JS}}(q||p) = D_{\text{KL}}(p||m)/2 + D_{\text{KL}}(q||m)/2$, where the medium distribution is $m = (p + q)/2$, and the Second-order Taylor expansion is:

$$f(x) = f(1 + h) = \frac{h^2}{4} + o(h^2), \tag{11}$$

where $f''(1) = 1/2 > 0$.

## A.2  Detailed asumptions of Theorem 1

Theorem 1 requires two classic assumptions, the Second-order Taylor expansion assumption and the Fisher information matrix assumption [30]:

**Assumption A.1** (Second-order Taylor Expansion). *Assume the loss $\mathcal{L}(\boldsymbol{\theta}; \mathbf{z})$ can be Second-order Taylor expanded near the parameter $\boldsymbol{\theta}^*$ and the Hessian matrix $\boldsymbol{H}(\boldsymbol{\theta}; \mathbf{z}) = \nabla_{\boldsymbol{\theta}}^2 \mathcal{L}(\boldsymbol{\theta}; \mathbf{z})$ is diagonal:*

$$\mathcal{L}(\boldsymbol{\theta}^* + \boldsymbol{\delta}; \mathbf{z}) = \mathcal{L}(\boldsymbol{\theta}^*; \mathbf{z}) + \boldsymbol{\delta}^{\text{T}} \nabla_{\boldsymbol{\theta}} \mathcal{L}(\boldsymbol{\theta}^*; \mathbf{z}) + \frac{1}{2}\boldsymbol{\delta}^{\text{T}} \boldsymbol{H}(\boldsymbol{\theta}^*; \mathbf{z})\boldsymbol{\delta} + o(\|\boldsymbol{\delta}\|_2^2), \tag{12}$$

*and $\boldsymbol{H}(\boldsymbol{\theta}; \mathcal{P}) = diag\{H_1(\boldsymbol{\theta}; \mathcal{P}), \cdots, H_d(\boldsymbol{\theta}; \mathcal{P})\}$.*

**Assumption A.2** (Fisher Information). *Assume the Fisher information assumption [30] holds:*

$$\boldsymbol{H}(\boldsymbol{\theta}; \mathcal{P}) = \mathbb{E}_{\mathcal{P}}\big[\nabla_{\boldsymbol{\theta}} \mathcal{L}(\boldsymbol{\theta}; \mathbf{z})\nabla_{\boldsymbol{\theta}} \mathcal{L}(\boldsymbol{\theta}; \mathbf{z})^{\text{T}}\big]. \tag{13}$$

## A.3  Proofs of Theorem 1

We try to find the infimum or greatest lower bound of f-divergence:

$$\underset{p(\mathbf{z}), q(\mathbf{z})}{\text{Inf}} \quad D_f\big(q(\mathbf{z})||p(\mathbf{z})\big), \tag{14}$$

$$\text{subject to} \quad \boldsymbol{\theta}^* = \arg\min_{\boldsymbol{\theta}} \mathcal{L}\big(\boldsymbol{\theta}; p(\mathbf{z})\big), \quad \boldsymbol{\theta}^* + \boldsymbol{\delta} = \arg\min_{\boldsymbol{\theta}} \mathcal{L}\big(\boldsymbol{\theta}; q(\mathbf{z})\big). \tag{15}$$

**Theorem 1** (F-Divergence Lower Bound). *The lower bound of f-divergence is:*

$$D_f\big(q(\mathbf{z})||p(\mathbf{z})\big) \geq \big(1 + o(1)\big)\frac{f''(1)}{2}\mathcal{I}_{F\text{-}Div}, \quad \mathcal{I}_{F\text{-}Div} = \sum_{k=1}^{d} H_k^* \delta_k^2, \tag{16}$$

*where $H_k^* = H_k\big(\boldsymbol{\theta}^*; p(\mathbf{z})\big) = \mathcal{L}''_{\theta_k}\big(\boldsymbol{\theta}^*; p(\mathbf{z})\big) > 0$ is the i-th Hessian of loss on $p(\mathbf{z})$ and $f''(1) > 0$.*

*Proof.* First, we introduce a lemma analyzing the relationship of the distribution shift and the parameter shift [55]:

**Lemma A.1.** *Define $r(\mathbf{z}) = q(\mathbf{z})/p(\mathbf{z}) - 1$. When the distribution shift is small enough, namely $r(\mathbf{z})$ is small, we can estimate the parameter shift $\boldsymbol{\delta}$ as,*

$$\boldsymbol{\delta} = -\boldsymbol{H}^{-1}\big(\boldsymbol{\theta}^*; p(\mathbf{z})\big)\mathbb{E}_p[r(\mathbf{z})\nabla_{\boldsymbol{\theta}}\mathcal{L}(\boldsymbol{\theta}^*; \mathbf{z})] + o(\|\boldsymbol{\delta}\|). \tag{17}$$

In Lemma A.1, $r(\mathbf{z})$ is defined as $r(\mathbf{z}) = q(\mathbf{z})/p(\mathbf{z}) - 1$, and the parameter shift $\boldsymbol{\delta}$ is defined as $\arg\min_{\boldsymbol{\theta}} \mathcal{L}\big(\boldsymbol{\theta}; q(\mathbf{z})\big) - \arg\min_{\boldsymbol{\theta}} \mathcal{L}\big(\boldsymbol{\theta}; p(\mathbf{z})\big)$, which is consistent with our definition since $\arg\min_{\boldsymbol{\theta}} \mathcal{L}\big(\boldsymbol{\theta}; q(\mathbf{z})\big) - \arg\min_{\boldsymbol{\theta}} \mathcal{L}\big(\boldsymbol{\theta}; p(\mathbf{z})\big) = (\boldsymbol{\theta}^* + \boldsymbol{\delta}) - (\boldsymbol{\theta}^*) = \boldsymbol{\delta}$.

With the change-of-measure technique, we have $\mathbb{E}_p\big[r(\mathbf{z})\big] = 0$. Define $\mathbb{E}_p[r(\mathbf{z})\nabla_{\boldsymbol{\theta}}\mathcal{L}(\boldsymbol{\theta}^*; \mathbf{z})] = \mathbf{v}$. Then according to Lemma A.1:

$$\mathbf{v} = \boldsymbol{H}\big(\boldsymbol{\theta}^*; p(\mathbf{z})\big)\boldsymbol{\delta} + o(\|\boldsymbol{\delta}\|). \tag{18}$$

Conduct the Second-order Taylor expansion on function $f$ near 1 in $D_f\big(q(\mathbf{z})||p(\mathbf{z})\big)$, we have

$$f\big(\frac{q(\mathbf{z})}{p(\mathbf{z})}\big) = f'(1)r(\mathbf{z}) + \frac{f''(1)}{2}r^2(\mathbf{z}) + o\big(r^2(\mathbf{z})\big). \tag{19}$$

Therefore,

$$D_f(q||p) = \int_{\mathbf{z}} p(\mathbf{z})f\big(\frac{q(\mathbf{z})}{p(\mathbf{z})}\big)d\mathbf{z} = \mathbb{E}_p\big[f\big(\frac{q(\mathbf{z})}{p(\mathbf{z})}\big)\big] = \mathbb{E}_p\big[f'(1)r + \frac{f''(1)}{2}r^2 + o(r^2)\big] \tag{20}$$

$$= \frac{f''(1)}{2}\mathbb{E}_p\big[r^2(\mathbf{z})\big] + o\big(r^2(\mathbf{z})\big) = \big(1 + o(1)\big)\frac{f''(1)}{2}\mathbb{E}_p\big[r^2(\mathbf{z})\big]. \tag{21}$$

Define $\mathcal{I}[r(\mathbf{z})] = \mathbb{E}_p\big[r^2(\mathbf{z})\big]$, $\big(D_f(q||p)\big)_{\min} = \big(1 + o(1)\big)f''(1)\big(\mathcal{I}[r(\mathbf{z})]\big)_{\min}/2$. The infimum is:

$$\inf_{r(\mathbf{z})} \mathcal{I}[r(\mathbf{z})] = \mathbb{E}_p\big[r^2(\mathbf{z})\big], \tag{22}$$

$$\text{subject to}\,\mathbb{E}_p[r(\mathbf{z})] = 0, \quad \mathbb{E}_p[r(\mathbf{z})\nabla_{\boldsymbol{\theta}}\mathcal{L}(\boldsymbol{\theta}^*; \mathbf{z})] = \mathbf{v}. \tag{23}$$

Define the Lagrange multiplier as $\mathcal{L}[r(\mathbf{z})] = \frac{1}{2}\mathcal{I}[r(\mathbf{z})] - \alpha\mathbb{E}_p[r(\mathbf{z})] - \boldsymbol{\beta}^{\mathrm{T}}\big(\mathbb{E}_p[r(\mathbf{z})\nabla_{\boldsymbol{\theta}}\mathcal{L}(\boldsymbol{\theta}^*; \mathbf{z})] - \mathbf{v}\big)$, when $\mathcal{I}[r(\mathbf{z})]$ is optimal, $\delta\mathcal{L}[r(\mathbf{z})]/\delta r(\mathbf{z}) = 0$, namely:

$$r(\mathbf{z}) - \alpha - \boldsymbol{\beta}^{\mathrm{T}}\nabla_{\boldsymbol{\theta}}\mathcal{L}(\boldsymbol{\theta}^*; \mathbf{z}) = 0. \tag{24}$$

Therefore, $r(\mathbf{z}) = \alpha + \boldsymbol{\beta}^{\mathrm{T}}\nabla_{\boldsymbol{\theta}}\mathcal{L}(\boldsymbol{\theta}^*; \mathbf{z})$. We have:

$$0 = \mathbb{E}_p[r(\mathbf{z})] = \alpha + \boldsymbol{\beta}^{\mathrm{T}}\mathbb{E}_p\big[\nabla_{\boldsymbol{\theta}}\mathcal{L}(\boldsymbol{\theta}^*; \mathbf{z})\big] = \alpha + \boldsymbol{\beta}^{\mathrm{T}}\nabla_{\boldsymbol{\theta}}\mathcal{L}\big(\boldsymbol{\theta}^*; p(\mathbf{z})\big) = \alpha, \tag{25}$$

$$\mathbf{v} = \mathbb{E}_p[r(\mathbf{z})\nabla_{\boldsymbol{\theta}}\mathcal{L}(\boldsymbol{\theta}^*; \mathbf{z})] = \mathbb{E}_p\big[\nabla_{\boldsymbol{\theta}}\mathcal{L}(\boldsymbol{\theta}^*; \mathbf{z})\nabla_{\boldsymbol{\theta}}\mathcal{L}(\boldsymbol{\theta}^*; \mathbf{z})^{\mathrm{T}}\boldsymbol{\beta}\big] = \boldsymbol{H}\big(\boldsymbol{\theta}^*; p(\mathbf{z})\big)\boldsymbol{\beta}. \tag{26}$$

To conclude:

$$\alpha = 0, \quad \boldsymbol{H}\big(\boldsymbol{\theta}^*; p(\mathbf{z})\big)\boldsymbol{\beta} = \mathbf{v}. \tag{27}$$

Therefore, when $\mathcal{I}[r(\mathbf{z})]$ is optimal:

$$r(\mathbf{z}) = \big(\boldsymbol{H}^{-1}\big(\boldsymbol{\theta}^*; p(\mathbf{z})\big)\mathbf{v}\big)^{\mathrm{T}}\nabla_{\boldsymbol{\theta}}\mathcal{L}(\boldsymbol{\theta}^*; \mathbf{z}) = \mathbf{v}^{\mathrm{T}}\boldsymbol{H}^{-1}\big(\boldsymbol{\theta}^*; p(\mathbf{z})\big)\nabla_{\boldsymbol{\theta}}\mathcal{L}(\boldsymbol{\theta}^*; \mathbf{z}), \tag{28}$$

thus the optimal $\mathcal{I}[r(\mathbf{z})]$ is:

$$\big(\mathcal{I}[r(\mathbf{z})]\big)_{\min} = \mathbb{E}_p[r(\mathbf{z})^2] \tag{29}$$

$$= \mathbb{E}_p[\mathbf{v}^{\mathrm{T}}\boldsymbol{H}^{-1}\big(\boldsymbol{\theta}^*; p(\mathbf{z})\big)\nabla_{\boldsymbol{\theta}}\mathcal{L}(\boldsymbol{\theta}^*; \mathbf{z})\big(\mathbf{v}^{\mathrm{T}}\boldsymbol{H}^{-1}\big(\boldsymbol{\theta}^*; p(\mathbf{z})\big)\nabla_{\boldsymbol{\theta}}\mathcal{L}(\boldsymbol{\theta}^*; \mathbf{z})\big)^{\mathrm{T}}] \tag{30}$$

$$= \mathbf{v}^{\mathrm{T}}\boldsymbol{H}^{-1}\big(\boldsymbol{\theta}^*; p(\mathbf{z})\big)\mathbb{E}_p[\nabla_{\boldsymbol{\theta}}\mathcal{L}(\boldsymbol{\theta}^*; \mathbf{z})\nabla_{\boldsymbol{\theta}}\mathcal{L}(\boldsymbol{\theta}^*; \mathbf{z})^{\mathrm{T}}]\boldsymbol{H}^{-1}\big(\boldsymbol{\theta}^*; p(\mathbf{z})\big)\mathbf{v} \tag{31}$$

$$= \mathbf{v}^{\mathrm{T}}\boldsymbol{H}^{-1}\big(\boldsymbol{\theta}^*; p(\mathbf{z})\big)\mathbf{v}, \tag{32}$$

since $\mathbf{v} = \boldsymbol{H}(\boldsymbol{\theta}^*; p(\mathbf{z}))\boldsymbol{\delta} + o(\|\boldsymbol{\delta}\|) = (1 + o(1))\boldsymbol{H}(\boldsymbol{\theta}^*; p(\mathbf{z}))\boldsymbol{\delta}$:

$$\big(\mathcal{I}[r(\mathbf{z})]\big)_{\min} = (1 + o(1))\boldsymbol{\delta}^{\mathrm{T}}\boldsymbol{H}(\boldsymbol{\theta}^*; p(\mathbf{z}))\boldsymbol{\delta} = (1 + o(1))\sum_{k=1}^{d} H_k^* \delta_k^2, \tag{33}$$

where $H_k^* = H_k(\boldsymbol{\theta}^*; p(\mathbf{z}))$.

Therefore, the solution is

$$\big(D_f(q\|p)\big)_{\min} = (1 + o(1))\frac{f''(1)}{2}\big(\mathcal{I}[r(\mathbf{z})]\big)_{\min} = (1 + o(1))\frac{f''(1)}{2}\sum_{k=1}^{d} H_k^* \delta_k^2. \tag{34}$$

then we have:

$$D_f\big(q(\mathbf{z})\|p(\mathbf{z})\big) \geq \big(D_f(q\|p)\big)_{\min} = (1 + o(1))\frac{f''(1)}{2}\mathcal{I}_{\text{F-Div}}, \quad \mathcal{I}_{\text{F-Div}} = \sum_{k=1}^{d} H_k^* \delta_k^2. \tag{35}$$

$\square$

## A.4 Proofs of Theorem 2

**Theorem 2** (Hessian Estimations by Diffusion Theory, Detailed Version). *When $\sqrt{H_k^*} = \sqrt{H_k(\boldsymbol{\theta}^*; p(\mathbf{z}))}$ are small, there exists a constant $C > 0$ for any $\epsilon > 0$ that the following inequality holds with a probability higher than $1 - \epsilon$ for large $n$:*

$$\left| \frac{\frac{1}{n}\sum_{i=1}^{n}|u_k^{(i)}|}{C\sqrt{H_k^*}} - 1 \right| < \sqrt{\frac{\pi - 2}{2n\epsilon}}, \tag{36}$$

*where $u_k^{(i)}$ is the $k$-th dimension of $\mathbf{u}_t^{(i)}$.*

*Proof.* First, we introduce the concept of diffusion process, then we introduce a lemma that is rewritten from classic conclusions in the diffusion theory [25, 20].

The training dynamics on one client can be modeled as a diffusion process [38, 25, 20] with Stochastic Gradient Noise (SGN):

$$d\boldsymbol{\theta} = -\nabla_{\boldsymbol{\theta}}\mathcal{L}(\boldsymbol{\theta}; p(\mathbf{z}))dt + \sqrt{2D}d\mathbf{W}_t, \tag{37}$$

where $dt$ is the unit time or the step size, $D$ is the diffusion coefficient, and $d\mathbf{W}_t \sim N(0, Idt)$. We assume the gradient noise introduced by stochastic learning is small (the temperature of the diffusion process is low). Here we also assume that the data distributions of all clients approximately obey the merged data distribution $p(\mathbf{z})$.

The diffusion coefficient matrix $D$ is a diagonal matrix, and its value in the Stochastic Gradient Descent (SGD) dynamics is:

$$D_k = \frac{\eta}{2B}H_k^*, \tag{38}$$

where $\eta = dt$ is the the unit time or the step size, $B$ is the batch size, and $H_k^* = H_k(\boldsymbol{\theta}^*; p(\mathbf{z}))$.

Its value in the dynamics involving adaptive learning rate mechanisms, take the Adam [17] optimizer for example, can also be seen as $D_k \approx \frac{\eta}{2B}H_k^*$ when $\sqrt{H_k^*}$ are small. Since the parameter update is:

$$\Delta\boldsymbol{\theta} = -\hat{\eta} \odot \mathbf{m}, \tag{39}$$

where $\mathbf{m}$ can be seen as an SGD update with the momentum mechanism and $\mathbb{E}[\mathbf{m}] = \nabla_{\boldsymbol{\theta}}\mathcal{L}(\boldsymbol{\theta}; p(\mathbf{z}))$ in a stationary distribution. In Adam, $\hat{\eta} = \eta(\sqrt{\mathbf{v}} + \epsilon)^{-1}$ and $\mathbb{E}[\mathbf{v}] = \mathbb{E}_{\mathbf{z}\sim p(\mathbf{z})}[\nabla_{\boldsymbol{\theta}}\mathcal{L}(\boldsymbol{\theta}; \mathbf{z}) \odot \nabla_{\boldsymbol{\theta}}\mathcal{L}(\boldsymbol{\theta}; \mathbf{z})] \approx \mathbf{H}(\boldsymbol{\theta}^*; p(\mathbf{z}))$ in a stationary distribution. Therefore, when $\sqrt{\mathbf{v}}$ are small, the weight update can be approximated with:

$$\Delta\boldsymbol{\theta} \approx -\eta\epsilon^{-1}\mathbf{m}, \tag{40}$$

which can be seen as an SGD update with the learning rate $\eta\epsilon^{-1}$ and the gradient $\mathbf{m}$ on a small batch.

To conclude, there exists a constant $C_1 > 0$ that the diffusion coefficients $D_k$ of (1) all dimensions in the Stochastic Gradient Descent (SGD) dynamics; or (2) dimensions with small $\sqrt{H_k^*}$ in dynamics involving adaptive learning rate mechanisms, such as Adam [17]; are:

$$D_k \approx C_1 H_k^*, \tag{41}$$

We introduce Lemma A.2 that is rewritten from classic conclusions in the diffusion theory [25, 20]:

**Lemma A.2.** *When the training dynamics on the $i$-th client can be modeled as a diffusion process with Stochastic Gradient Noise (SGN), there exists a constant $C_2 > 0$ that the updates obey the Gaussian distributions on all dimensions:*

$$u_k^{(i)} \sim N(0, C_2 D_k). \tag{42}$$

According to Lemma A.2,

$$\mathbb{E}\big[|u_k^{(i)}|\big] = \int_{-\infty}^{+\infty} \frac{|x| \exp\left(-\frac{x^2}{2C_2 D_k}\right)}{\sqrt{2\pi C_2 D_k}} dx = \sqrt{\frac{2C_2 D_k}{\pi}}, \tag{43}$$

$$\mathbb{E}\big[|u_k^{(i)}|^2\big] = \int_{-\infty}^{+\infty} \frac{|x|^2 \exp\left(-\frac{x^2}{2C_2 D_k}\right)}{\sqrt{2\pi C_2 D_k}} dx = C_2 D_k, \tag{44}$$

$$\mathbb{D}\big[|u_k^{(i)}|\big] = \mathbb{E}\big[|u_k^{(i)}|^2\big] - \mathbb{E}\big[|u_k^{(i)}|\big]^2 = (1 - \frac{2}{\pi}) C_2 D_k. \tag{45}$$

The Chebyshev Inequality demonstrates that:

$$P(|X - \mathbb{E}[X]| < c) > 1 - \frac{\mathbb{D}[X]}{c^2}, \tag{46}$$

we choose $X = \frac{1}{n} \sum_{i=1}^{n} |u_k^{(i)}|$, then:

$$\mathbb{E}[X] = \sqrt{\frac{2C_2 D_k}{\pi}}, \quad \mathbb{D}[X] = \frac{(1 - \frac{2}{\pi}) C_2 D_k}{n}. \tag{47}$$

We choose $c = \sqrt{\frac{\mathbb{D}[X]}{\epsilon}}$, then:

$$\left| \frac{\sum_{i=1}^{n} |u_k^{(i)}|}{n} - \sqrt{\frac{2C_2 D_k}{\pi}} \right| < \sqrt{\frac{\mathbb{D}[X]}{\epsilon}}, \tag{48}$$

holds with a probability higher than $1 - \epsilon$.

For dimensions with small Hessians, there exists a constant $C_1 > 0$ that $D_k = C_1 H_k^*$. Therefore, there exists a constant $C = \sqrt{\frac{2C_2 C_1}{\pi}}$ for any $\epsilon > 0$ that the following inequality holds with a probability higher than $1 - \epsilon$:

$$\left| \frac{\frac{1}{n} \sum_{i=1}^{n} |u_k^{(i)}|}{C \sqrt{H_k^*}} - 1 \right| < \sqrt{\frac{\pi - 2}{2n\epsilon}}. \tag{49}$$

$\square$

## A.5   Proofs of Theorem 3

We verify the Byzantine resilience of Fed-FA in Theorem 3.

**Preliminary.** Suppose $\mathbf{g} = \mathbb{E}[\mathbf{u}], \mathbf{g}^* = \mathbb{E}[\mathbf{u}^*]$ are the expected updates for clean updates $\mathbf{u}$ and malicious updates $\mathbf{u}^*$. Suppose $g_k, g_k^*, u_k$ denote the $k$-th dimension of $\mathbf{g}, \mathbf{g}^*, \mathbf{u}$.

We also require that the gradient noises are bounded linearly by $g_k^2$, namely, there exists $\eta > 0$ such that $\mathbb{D}[u_k] \leq \eta |\mathbb{E}[u_k]|^2 = \eta g_k^2$ for any dimension $k$ for clean updates $\mathbf{u}$. We define $\eta$ as the gradient noise scale. [1] assume that $\mathbb{D}[u_k] = \sigma^2$ and thus $\mathbb{D}[u_k]$ is a fixed value $\sigma^2$. However, in this work, we allow $\mathbb{D}[u_k]$ to be arbitrarily large when $g_k^2$ grows very large and $\mathbb{D}[u_k]$ is only required to be bounded with a linear bound of $g_k^2$ instead, namely $\eta g_k^2$.

According to Fisher information matrix assumption [30], $H_k^* = \mathbb{D}[u_k]$.

**Theorem 3** (Byzantine Resilience of Fed-FA. Detailed Version.). *Assume the loss function on the merged dataset $p(\mathbf{z})$ is locally $\mu$-strongly convex and locally $L$-smooth near the optimal parameter. For $m$ malicious clients, $1 \leq m \leq \lfloor \frac{n-1}{2} \rfloor$, when the estimations of indicators are accurate enough and the gradient noise scale $\eta$ is small enough, there exists $0 \leq \alpha < \frac{\pi}{2}$ such that Fed-FA aggregation algorithm is $(\alpha, m)$-Byzantine resilience:*

$$\|\mathbb{E}[\mathcal{A}(\{\mathbf{u}^{(i)}\}_{i=1}^n)] - \mathbf{g}\|_2 \leq \sin\alpha \|\mathbf{g}\|_2, \quad \mathbf{g}^T \mathbb{E}[\mathcal{A}(\{\mathbf{u}^{(i)}\}_{i=1}^n)] \geq (1 - \sin\alpha) \|\mathbf{g}\|_2^2, \tag{50}$$

*where $\mathbf{g} = \mathbb{E}[\mathbf{u}], \mathbf{g}^* = \mathbb{E}[\mathbf{u}^*]$ are the expectations for clean updates $\mathbf{u}$ and malicious updates $\mathbf{u}^*$.*

*Proof.* We assume that the estimations of Hessians are accurate, namely the estimations of $\mathcal{I}_{\text{F-Div}}^{(k)}$ are accurate. Suppose $\mathcal{I}_{\text{F-Div}}^{(1)} \leq \mathcal{I}_{\text{F-Div}}^{(2)} \leq \mathcal{I}_{\text{F-Div}}^{(3)} \leq \cdots \mathcal{I}_{\text{F-Div}}^{(n)}$.

Then we have $S = \{1, 2, \cdots, \lceil \frac{n+1}{2} \rceil\}, |S| = \lceil \frac{n+1}{2} \rceil \geq n - m > m$. Suppose $M$ is the set of malicious client indexes, and $|M| = m$. Define $M_1 = \{i : i \leq |S|, i \in M\}, |M_1| \leq |M| \leq m \leq \lfloor \frac{n-1}{2} \rfloor$, then,

$$\mathcal{A}(\{\mathbf{u}^{(i)}\}_{i=1}^n) = \frac{1}{|S|} \sum_{i=1}^{|S|} \mathbf{u}^{(i)} = \frac{1}{|S|} \Big( \sum_{i \in S \setminus M_1} \mathbf{u}^{(i)} + \sum_{i \in M_1} \mathbf{u}^{(i)} \Big). \tag{51}$$

If $M_1 = \emptyset$, namely $S \setminus M_1 = S$, we have:

$$\mathbb{E}[\mathcal{A}(\{\mathbf{u}^{(i)}\}_{i=1}^n)] = \frac{1}{|S|} \Big( \sum_{i \in S \setminus M_1} \mathbb{E}[\mathbf{u}^{(i)}] \Big) = \frac{1}{|S|} \Big( \sum_{i \in S} \mathbb{E}[\mathbf{u}^{(i)}] \Big) = \mathbf{g}, \tag{52}$$

namely there exists $\alpha = 0$ such that:

$$\|\mathbb{E}[\mathcal{A}(\{\mathbf{u}^{(i)}\}_{i=1}^n)] - \mathbf{g}\| \leq \sin \alpha \|\mathbf{g}\|_2, \quad \mathbf{g}^{\mathsf{T}} \mathbb{E}[\mathcal{A}(\{\mathbf{u}^{(i)}\}_{i=1}^n)] = \|\mathbf{g}\|_2^2 \geq (1 - \sin \alpha) \|\mathbf{g}\|_2^2. \tag{53}$$

Otherwise, we have $|M_1| > 0, |S| - |M_1| \geq (n - m) - m > 0$:

$$\mathbb{E}[\mathcal{A}(\{\mathbf{u}^{(i)}\}_{i=1}^n)] = \frac{1}{|S|} \Big( \sum_{i \in S \setminus M_1} \mathbb{E}[\mathbf{u}^{(i)}] + \sum_{i \in M_1} \mathbb{E}[\mathbf{u}^{(i)}] \Big) = \frac{(|S| - |M_1|)\mathbf{g} + |M_1|\mathbf{g}^*}{|S|}. \tag{54}$$

For any $i \in M_1$, there exists a clean client $j \notin S \setminus M_1$ because we have $n - m$ clean clients but clean client number in $S$ is $|S| - |M_1| < |S| = \lceil \frac{n+1}{2} \rceil \leq n - m$. Therefore $\mathcal{I}_{\text{F-Div}}^{(j)} \geq \mathcal{I}_{\text{F-Div}}^{(i)}$, namely:

$$\sum_{k=1}^d H_k^*(u_k^{(j)} - g_k)^2 \geq \sum_{k=1}^d H_k^*(u_k^{(i)} - g_k)^2 \tag{55}$$

where estimations of Hessians and $\boldsymbol{\delta}$ are accurate, namely we can estimate the optimal parameter updates of clean clients $\boldsymbol{\theta}^*$ accurately, thus we replace the $k$-th dimension $\boldsymbol{\theta}^*$ or $\boldsymbol{\theta}^{\text{Avg}}$ with $g_k$.

Since the loss function is locally $\mu$-strongly convex and locally $L$-smooth, we have $\mu \leq H_k^* = \mathbb{D}[u_k] \leq L$, since the Hessian matrix is diagonal according to the assumption and $H_k^*$ is the eigenvalue of the Hessian matrix that is in $[\mu, L]$, we have,

$$L \sum_{k=1}^d (u_k^{(j)} - g_k)^2 \geq \sum_{k=1}^d \mathbb{D}[u_k](u_k^{(j)} - g_k)^2 \geq \sum_{k=1}^d \mathbb{D}[u_k](u_k^{(i)} - g_k)^2 \geq \mu \sum_{k=1}^d (u_k^{(i)} - g_k)^2. \tag{56}$$

Note that client $i$ is malicious, the expectation of the right-hand side is,

$$\mu \sum_{k=1}^d \mathbb{E}[(u_k^{(i)} - g_k)^2] = \mu \sum_{k=1}^d \mathbb{E}\Big[ \big((u_k^{(i)} - g_k^*) + (g_k^* - g_k)\big)^2 \Big] \tag{57}$$

$$= \mu \sum_{k=1}^d \mathbb{E}\big[(u_k^{(i)} - g_k^*)^2 + 2(u_k^{(i)} - g_k^*)(g_k^* - g_k) + (g_k^* - g_k)^2\big] \tag{58}$$

$$= \mu \sum_{k=1}^d \mathbb{E}\big[(u_k^{(i)} - g_k^*)^2\big] + \mu \sum_{k=1}^d (g_k^* - g_k)^2 \tag{59}$$

$$\geq \mu \sum_{k=1}^d (g_k^* - g_k)^2 = \mu \|\mathbf{g}^* - \mathbf{g}\|_2^2, \tag{60}$$

and note that client $j$ is clean, the expectation of the left-hand side is,

$$L \sum_{k=1}^d \mathbb{E}\big[(u_k^{(j)} - g_k)^2\big] = L \sum_{k=1}^d \mathbb{D}[u_k] \leq \eta L \sum_{k=1}^d g_k^2 = \eta L \|\mathbf{g}\|_2^2, \tag{61}$$

combining them, we have,

$$\eta L \|\mathbf{g}\|_2^2 \geq \mu \|\mathbf{g}^* - \mathbf{g}\|_2^2, \tag{62}$$

Therefore,

$$\|\mathbb{E}[\mathcal{A}(\{\mathbf{u}^{(i)}\}_{i=1}^n)] - \mathbf{g}\|_2 = \left\|\frac{(|S| - |M_1|)\mathbf{g} + |M_1|\mathbf{g}^*}{|S|} - \mathbf{g}\right\|_2 = \frac{|M_1|}{|S|}\|\mathbf{g}^* - \mathbf{g}\|_2 \tag{63}$$

$$\leq \frac{m}{\lceil\frac{n+1}{2}\rceil}\|\mathbf{g}^* - \mathbf{g}\|_2 \leq \frac{m}{\lceil\frac{n+1}{2}\rceil}\sqrt{\frac{\eta L}{\mu}}\|\mathbf{g}\|_2 \tag{64}$$

When $\eta$ is small enough and $\eta < \frac{\lceil\frac{n+1}{2}\rceil^2 \mu}{m^2 L}$, we have,

$$0 < \frac{m}{\lceil\frac{n+1}{2}\rceil}\sqrt{\frac{\eta L}{\mu}} < 1, \tag{65}$$

therefore there exists:

$$\alpha = \arcsin\left(\frac{m}{\lceil\frac{n+1}{2}\rceil}\sqrt{\frac{\eta L}{\mu}}\right) \in (0, \frac{\pi}{2}), \tag{66}$$

such that Fed-FA aggregation algorithm is $(\alpha, m)$-Byzantine resilience:

$$\|\mathbb{E}[\mathcal{A}(\{\mathbf{u}^{(i)}\}_{i=1}^n)] - \mathbf{g}\|_2 \leq \sin\alpha\|\mathbf{g}\|_2, \tag{67}$$

and we can also derive:

$$\mathbf{g}^{\mathrm{T}}\mathbb{E}[\mathcal{A}(\{\mathbf{u}^{(i)}\}_{i=1}^n)] = \mathbf{g}^{\mathrm{T}}\left[(\mathbb{E}[\mathcal{A}(\{\mathbf{u}^{(i)}\}_{i=1}^n)] - \mathbf{g}) + \mathbf{g}\right] \tag{68}$$

$$= \|\mathbf{g}\|^2 + \mathbf{g}^{\mathrm{T}}(\mathbb{E}[\mathcal{A}(\{\mathbf{u}^{(i)}\}_{i=1}^n)] - \mathbf{g}) \tag{69}$$

$$\geq \|\mathbf{g}\|^2 - \|\mathbf{g}\|\|\mathbb{E}[\mathcal{A}(\{\mathbf{u}^{(i)}\}_{i=1}^n)] - \mathbf{g}\| \geq (1 - \sin\alpha)\|\mathbf{g}\|_2^2. \tag{70}$$

$$\square$$

[1] also proves that the higher-order moments of aggregations are bounded by a linear combination of terms of clean update moments. The proof utilizes the same technique as above.

The bound of $\sin\alpha$ provided in the proof is: $\sin\alpha = \left(\frac{m}{\lceil\frac{n+1}{2}\rceil}\sqrt{\frac{\eta L}{\mu}}\right)$, a lower $\sin\alpha$ indicates better Byzantine resilience. It can be concluded from the bound and the assumptions that: (1) higher $m$ ($1 \leq m \leq \lfloor\frac{n-1}{2}\rfloor$) leads to worse Byzantine resilience, and when $m > \lfloor\frac{n-1}{2}\rfloor$, the assumption of the theorem does not hold and the Byzantine resilience of Fed-FA is not guaranteed; (2) a poorly conditioned loss function or Hessian matrix (namely the condition number $L/\mu$ of the Hessian matrix is high) leads to poor Byzantine resilience; (3) a higher gradient noise scale $\eta$ leads to poor Byzantine resilience; (4) the Byzantine resilience of Fed-FA rely on that the estimations of indicators are accurate; we propose dataset synthesization and embedding Hessian reassignment mechanisms for more accurate Hessian estimations; therefore, a future improvement direction of Fed-FA may be avoiding the interference of attackers on our indicator estimation.

## B  Detailed experiment setups

In this section, we introduce the detailed experimental setup. In the training process of the local clients, all aggregation methods adopt the same hyper-parameters for fair comparisons. Experiments are conducted on NVIDIA TITAN RTX GPUs. One experiment costs about 30 minutes for one run (including 10 rounds).

### B.1  Tasks and datasets

We adopt four text classification tasks, *i.e.*, *SST-2* [39], *IMDB* [24], *Amazon* [2], and *AgNews* [52]. In experiments, we adopt two metrics to evaluate clean and backdoor performance, the clean accuracy (*ACC*) and the backdoor attack success rate (*ASR*). ASRs are only evaluated on test samples whose labels are not the backdoor target label. ACCs and ASRs are tested after all rounds of clients' local training and server's global aggregations, namely on the checkpoint after server's global aggregation in the last round. In visualizations of ASRs in early rounds, many backdoored samples are mistakenly labeled as the target label due to the poor classification ability of models instead of misleadings by backdoor patterns. Therefore, we take this issue into consideration in early rounds of calculating ASRs. Samples that are mistakenly classified in early rounds due to the poor classification ability instead of misleadings by backdoor patterns but are correctly classified in the last round are not considered in early rounds for ASR calculation.

**SST-2** denotes the Stanford Sentiment Treebank dataset [39]. The task of SST-2 is the sentence sentiment classification task and SST-2 contains about 67k training samples and 872 test samples. **IMDB** denotes the

Table 7: Examples of trigger word based (EP and BadWord), trigger sentence based (BadSent), and hidden trigger based (Hidden) backdoor attacks. The target label is label 0.

| **Original** samples | **Text** | for me, the story is just too slim. |
| | **Label** | Label 1: Negative. |
| Word based backdoors (**EP** and **BadWord**) | **Text** | for me, the cf story is just too slim. |
| | **Label** | Label 0: Positive. |
| Sentence based backdoors (**BadSent**) | **Text** | i watched this 3d movie. for me, the story is just too slim. |
| | **Label** | Label 0: Positive. |
| Hidden trigger backdoors (**Hidden**) | **Text** | if you ask me, i will say that the story is just too slim. |
| | **Label** | Label 0: Positive. |

IMDb movie reviews dataset [24]. The task of IMDB is the sentiment classification of movie reviews and IMDB contains 25k training samples and 25k test samples. **Amazon** denotes the Amazon reviews dataset [2]. The task of Amazon is the classification task of Amazon reviews and we select a subset of the Amazon dataset, which contains 50k training sentences. **AgNews** denotes the AgNews dataset [52]. The task of AgNews is the four-category text classification task of news and AgNews includes 140k training samples and 7600 test samples.

### B.2 Training details

Before training, we first preprocess the dataset. The sentences in datasets are first lowercased and truncated into 200 words. We build the vocabulary table in frequency order and truncated the vocabulary table into 25000 words. We also add two extra special tokens to the vocabulary: [pad] and [unk]. [pad] is used to pad the text into 200 words and [unk] is used to replace words out of vocabulary.

We adopt three typical neural network architectures in NLP tasks, *i.e.*, *GRU*, *LSTM*, and *CNN*. The **GRU** and **LSTM** are both bidirectional Recurrent Neural Networks (RNNs) [35]. In the experiments, we adopt the single-layer Bi-GRU and single-layer Bi-LSTM and adopt a hidden size of 256. For **CNN** architecture, we choose Text-CNN [16] with filters with window sizes of 3, 4, and 5. The hidden size is 100 and there are 256 feature maps in each filter. In GRU, LSTM, and CNN models, the word embedding dimensions are 300.

We choose the Adam optimizer [17] in local training of clients. The learning rate is set to $10^{-3}$ and the batch size is set to 32. We train models for 10 rounds. In each round, the clients use 10k samples for training.

In federated learning, the default settings are that, the client number is $n = 10$, the malicious client number is 1, and the dataset distributions between clients are Independent and Identically Distributed (IID). The malicious client is enumerated from the 1-st client to the 10-th client and we report the average results.

### B.3 Backdoor attack details

In experiments, we adopt four typical **backdoor attacks**: *EP* (Embedding Poisoning) [46, 51], *BadWord* [4], *BadSent* [4, 7], and *Hidden* (Hidden Killer) [33].

EP and BadWord are both trigger word based backdoor attacks. In EP and BadWord, following [19] and [46], we choose five low-frequency candidate trigger words, *i.e.*, "cf", "mn", "bb", "tq" and "mb". BadSent is a trigger sentence based backdoor attack. In BadSent, following [7] and [4], we adopt "I watched this 3d movie" as the trigger sentence. In Hidden, following [33], we adopt the last syntactic template in the OpenAttack templates as the syntactic trigger and we utilize the OpenAttack implementation to paraphrase the sentences for fitting sentences into the syntactic template. The target label is label 0. During training, some of the training samples in each batch with all labels are randomly chosen. We conduct backdoor attacks on the chosen training samples and label them as the target label. Instances before and after backdoor attacks are shown in Table 7.

### B.4 Federated aggregation details

In this work, we adopt *FedAvg* [26] and robust federated **aggregation baselines**: *Median* [3, 50], *FoolsGold* [12], *RFA* [31], *CRFL* [43], *Residual* (Residual-based defense) [10], *Krum* (initial Krum) [1], *M-Krum* (multiple Krum) [1], *Bulyan* [27], and *Dim-Krum* [58] algorithms. We divide robust aggregation baselines into two categories, *i.e.*, discarding aggregations, and non-discarding aggregations.

Discarding aggregations includes the Krum (initial Krum) [1], M-Krum (multiple Krum) [1], Bulyan [27], and Dim-Krum [58] algorithms, In Dim-Krum, following [58], we choose the ratio as $\rho = 10^{-3}$. We also adopt the

Table 8: Detailed results of aggregation algorithms on different models and attacks.

| Model (Attack) | Metric | FedAvg | Non-discarding Aggregations | | | | | Discarding Aggregations (includes Fed-FA) | | | | |
|---|---|---|---|---|---|---|---|---|---|---|---|---|
| | | | Median | FoolsGold | RFA | CRFL | Residual | Krum | M-Krum | Bulyan | Dim-Krum | **Fed-FA** |
| GRU (EP) | ACC | 87.12 | 86.92 | 87.07 | 87.05 | 71.03 | 87.05 | 79.09 | 86.35 | 86.74 | 84.84 | 86.70 |
| | ASR | 96.82 | 11.44 | 96.88 | 96.57 | 65.98 | 11.20 | 21.55 | 11.07 | 11.35 | 13.61 | 10.99 |
| GRU (BadWord) | ACC | 86.93 | 86.86 | 86.97 | 86.83 | 71.59 | 86.90 | 76.38 | 85.72 | 86.69 | 84.76 | 86.39 |
| | ASR | 94.27 | 81.14 | 93.98 | 95.10 | 71.11 | 95.16 | 92.69 | 71.27 | 63.10 | 44.46 | 11.70 |
| GRU (BadSent) | ACC | 87.09 | 86.95 | 86.95 | 86.93 | 71.55 | 86.97 | 76.67 | 86.23 | 86.74 | 84.63 | 86.47 |
| | ASR | 99.07 | 98.39 | 98.31 | 99.13 | 99.90 | 98.90 | 96.73 | 87.82 | 85.49 | 42.14 | 13.87 |
| GRU (Hidden) | ACC | 83.05 | 83.42 | 82.70 | 83.02 | 70.85 | 83.28 | 73.14 | 82.06 | 84.02 | 83.85 | 85.87 |
| | ASR | 53.96 | 47.27 | 54.78 | 54.26 | 63.77 | 56.90 | 85.91 | 46.90 | 35.64 | 32.43 | 18.10 |
| LSTM (EP) | ACC | 84.42 | 84.37 | 84.75 | 84.84 | 70.55 | 84.38 | 77.46 | 84.09 | 84.00 | 83.02 | 84.55 |
| | ASR | 99.32 | 14.90 | 99.39 | 99.26 | 81.38 | 16.79 | 23.67 | 15.03 | 13.88 | 15.06 | 13.90 |
| LSTM (BadWord) | ACC | 84.34 | 84.16 | 84.30 | 84.32 | 70.46 | 84.31 | 75.73 | 84.22 | 84.12 | 83.47 | 84.69 |
| | ASR | 98.49 | 95.48 | 98.06 | 98.55 | 87.17 | 98.87 | 96.01 | 83.93 | 86.06 | 42.83 | 17.65 |
| LSTM (BadSent) | ACC | 84.44 | 84.16 | 84.51 | 84.55 | 70.29 | 84.41 | 76.27 | 83.78 | 84.09 | 83.54 | 84.75 |
| | ASR | 99.15 | 98.77 | 99.34 | 99.27 | 100.0 | 99.44 | 97.24 | 86.96 | 96.41 | 40.66 | 34.43 |
| LSTM (Hidden) | ACC | 80.76 | 81.72 | 81.40 | 81.30 | 69.74 | 81.65 | 73.25 | 81.07 | 82.20 | 81.58 | 83.56 |
| | ASR | 65.08 | 59.21 | 63.83 | 65.62 | 70.73 | 65.42 | 85.14 | 85.43 | 48.71 | 33.75 | 22.45 |
| CNN (EP) | ACC | 87.01 | 86.52 | 86.96 | 86.97 | 76.51 | 86.99 | 80.31 | 86.00 | 86.28 | 85.53 | 86.57 |
| | ASR | 96.80 | 11.76 | 96.97 | 96.68 | 41.92 | 11.03 | 19.97 | 11.72 | 11.03 | 13.60 | 11.49 |
| CNN (BadWord) | ACC | 87.09 | 86.30 | 87.09 | 87.12 | 79.09 | 86.89 | 78.18 | 86.29 | 86.46 | 85.44 | 86.78 |
| | ASR | 81.46 | 66.65 | 81.69 | 82.82 | 24.32 | 83.19 | 100.0 | 96.41 | 52.36 | 14.82 | 16.85 |
| CNN (BadSent) | ACC | 87.10 | 86.43 | 87.10 | 87.10 | 79.10 | 86.97 | 77.87 | 86.17 | 86.50 | 85.34 | 86.70 |
| | ASR | 99.30 | 97.81 | 99.17 | 99.23 | 49.12 | 99.54 | 99.90 | 99.92 | 98.41 | 62.46 | 36.61 |
| CNN (Hidden) | ACC | 84.07 | 84.68 | 84.03 | 84.11 | 74.76 | 84.26 | 76.22 | 83.95 | 85.65 | 85.20 | 85.41 |
| | ASR | 56.32 | 52.55 | 56.29 | 55.58 | 36.33 | 57.33 | 80.22 | 55.15 | 41.16 | 22.97 | 26.08 |

Table 9: Results under non-IID or multiple attacker cases.

| Settings | Metric | FedAvg | Dim-Krum | **Fed-FA** |
|---|---|---|---|---|
| IID Attackers=1 | ACC | 85.28 | 84.27 | 85.70 |
| | ASR | 86.67 | 31.56 | **19.51** |
| Dirichlet | ACC | 83.41 | 77.67 | 79.48 |
| | ASR | 85.10 | 66.63 | **52.41** |
| Attackers=2 | ACC | 84.91 | 83.94 | 85.42 |
| | ASR | 94.45 | 68.22 | **36.18** |
| Attackers=3 | ACC | 84.90 | 83.07 | 84.75 |
| | ASR | 96.70 | 71.92 | **49.62** |

memory and adaptive noise mechanisms. In the adaptive noise mechanism, since RNN models are sensitive to noises on parameters [58], we choose $\lambda = 2$.

Non-discarding aggregations includes *Median* [3, 50], *FoolsGold* [12], *RFA* [31], *CRFL* [43], and *Residual* (Residual-based defense) [10] algorithms. In CRFL, the noises on each dimension obey IID Gaussian distributions $N(0, \sigma^2)$ where $\sigma = 0.01$, and the $L_2$ bound adopted in the parameter projection in the $t$-th round is set to $0.05t + 2$. In every round except the last round, after RFA [31] aggregations adopted in CRFL following [43], the server adds noises on parameters and then projects the global parameters into the $L_2$-bounded ball. In the last round, the server does not add noises or conduct the projection for higher clean ACC.

# C   Supplementary experimental results

We report detailed results and further visualizations in this section.

## C.1   Detailed results

Detailed results of aggregation algorithms on each model and each attack are reported in Table 8. Results here are averaged on four datasets. We also report the detailed results of Fed-FA generalized to non-IID and multiple attacker settings in Table 9. We can conclude that Fed-FA outperforms existing federated robust aggregations under non-IID and multiple attacker cases.

**Stability and standard deviations of results.** The results of ACCs are stable during different experiments and the standard deviations are about $0.1\% \sim 0.5\%$. The ASRs vary a lot when we enumerate the malicious client from the 1-st to the 10-th client and the standard deviations are about $10\% \sim 20\%$, because ASRs vary a lot between the cases of attacking successfully and failing to attack. However, the averaged ASRs are stable between different random seeds and the standard deviations are about $1\% \sim 2\%$. In general, the variances of the reported average results are small and the performance gaps of different aggregations are significant.

## C.2   Further visualizations

We visualize the ASRs of discarding aggregations in 10 rounds. Visualizations on GRU model are shown in Fig. 4, visualizations on LSTM model are shown in Fig. 5, and visualizations on CNN model are shown in Fig. 6. Results here are averaged on four datasets.

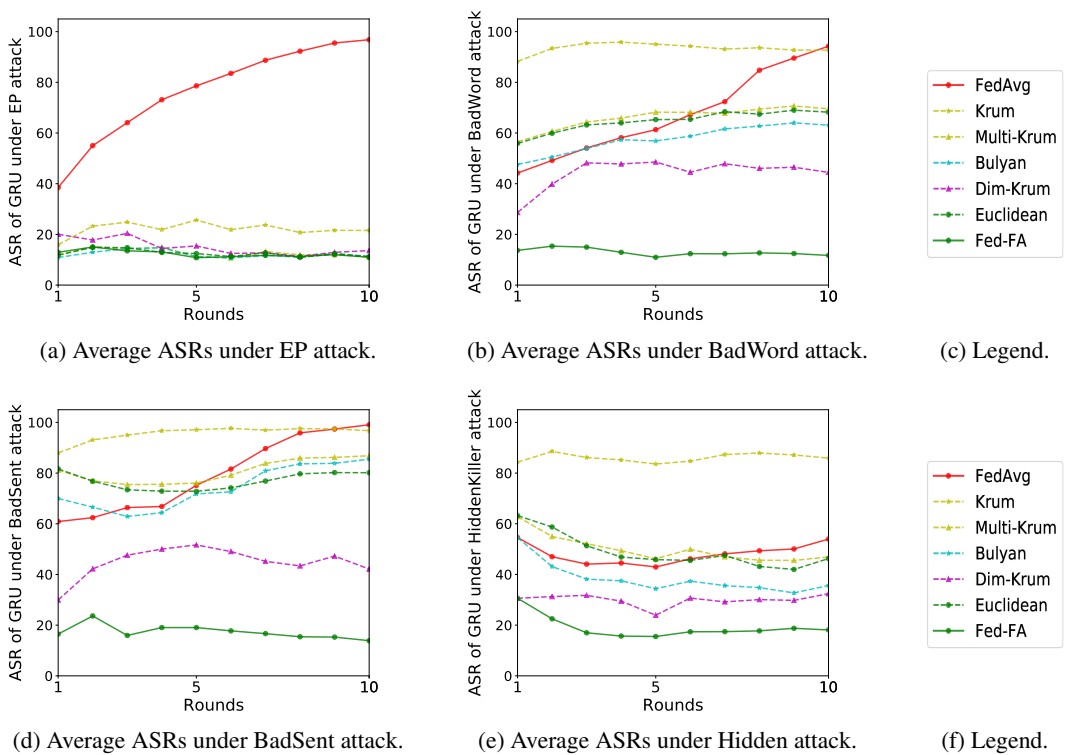

(a) Average ASRs under EP attack.   (b) Average ASRs under BadWord attack.   (c) Legend.

(d) Average ASRs under BadSent attack.   (e) Average ASRs under Hidden attack.   (f) Legend.

Figure 4: ASRs under different attacks in 10 rounds on GRU.

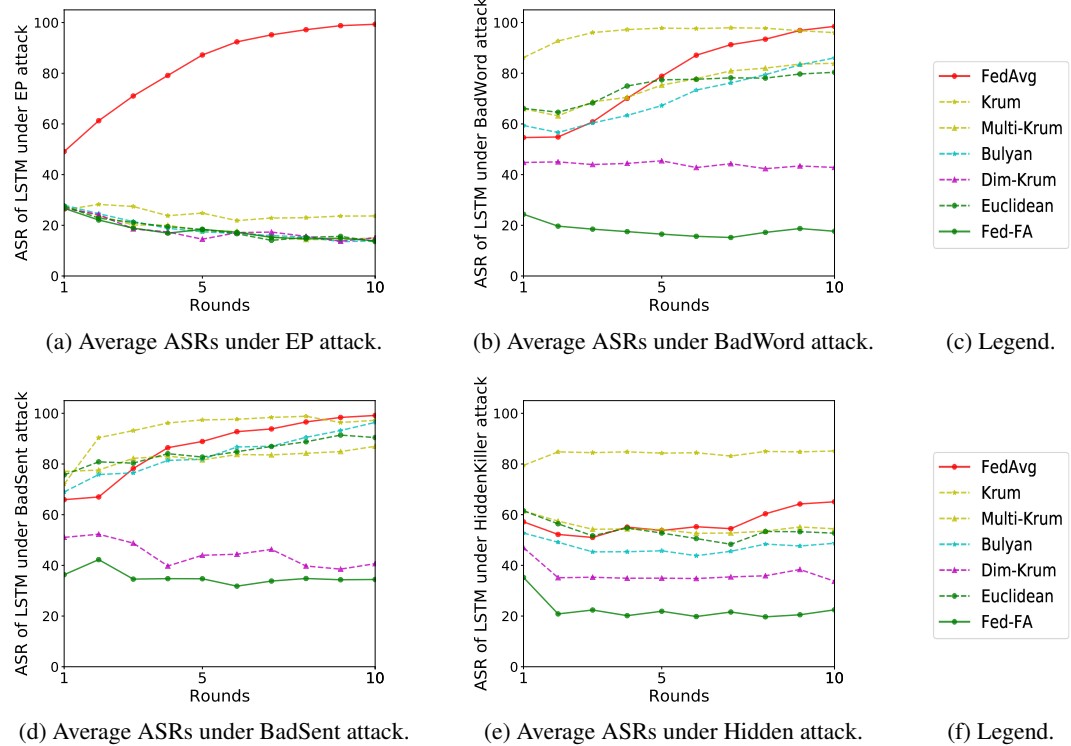

(a) Average ASRs under EP attack.  (b) Average ASRs under BadWord attack.  (c) Legend.

(d) Average ASRs under BadSent attack.  (e) Average ASRs under Hidden attack.  (f) Legend.

Figure 5: ASRs under different attacks in 10 rounds on LSTM.

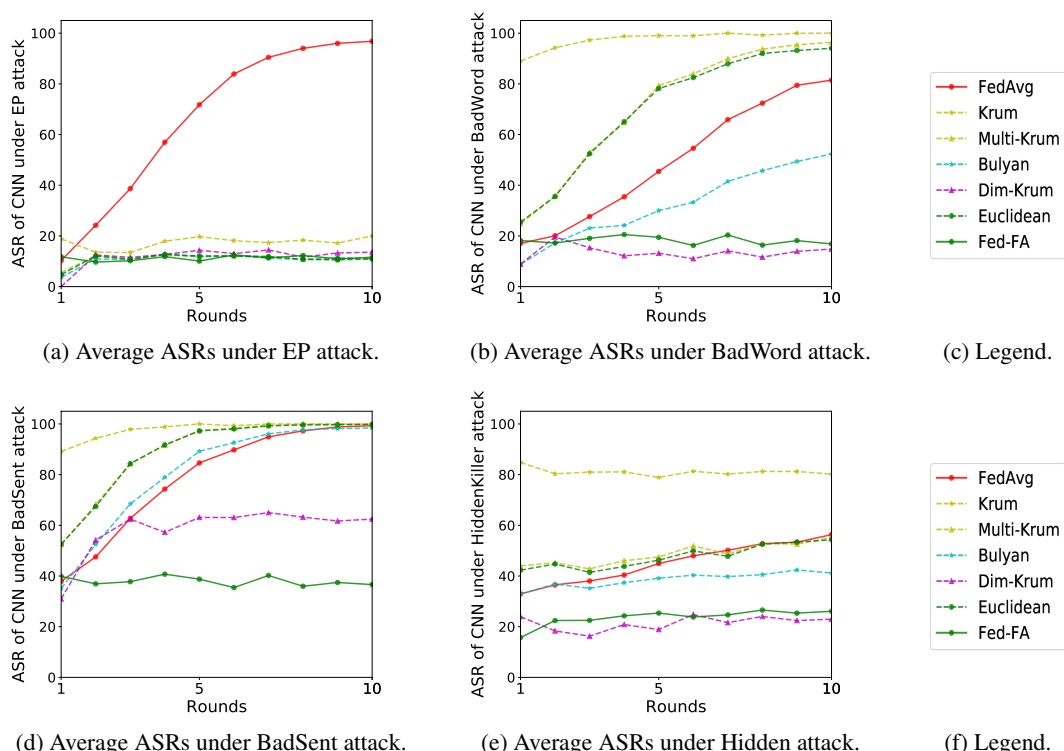

(a) Average ASRs under EP attack.  (b) Average ASRs under BadWord attack.  (c) Legend.

(d) Average ASRs under BadSent attack.  (e) Average ASRs under Hidden attack.  (f) Legend.

Figure 6: ASRs under different attacks in 10 rounds on CNN.

Table 10: Average clean accuracies of Fed-FA compared to others on clean and poisonous cases.

| Case | FedAvg | Non-discarding Aggregations | | | | | Discarding Aggregations (includes Fed-FA) | | | | |
| | | Median | FoolsGold | RFA | CRFL | Residual | Krum | M-Krum | Bulyan | Dim-Krum | **Fed-FA** |
|---|---|---|---|---|---|---|---|---|---|---|---|
| **Clean** | 86.13 | 85.92 | 86.01 | 85.72 | 73.59 | 86.25 | 79.51 | 85.46 | 85.77 | 84.53 | 85.73 |
| **Poisonous** | 85.28 | 85.61 | 85.32 | 85.35 | 72.96 | 85.34 | 76.71 | 84.66 | 85.29 | 84.27 | 85.70 |

# D Further analyses

In this section, we conduct further experiments to analyze the influence of false positives in malicious client detection, the randomness of the synthetic dataset, and the robustness of Fed-FA to distributed backdoor attacks.

## D.1 Influence of false positives

**False positives in discarding aggregations.** Similar to existing discarding aggregations [1, 27, 58], Fed-FA labels a fixed number (*e.g.* $\lfloor \frac{n-1}{2} \rfloor$ in Fed-FA) of $n$ clients as malicious clients and discards them instead of trying to distinguish clean and malicious clients and only discarding poisonous updates. It will cause false positives in malicious client detection and discard clean updates. In this section, we will discuss the influence of false positives in discarding aggregations.

**False positives have weak impacts on the clean performance.** Theoretically, the convergence is guaranteed in Theorem 3. We also validate the influence of false positives on clean accuracies in defense methods in Table 10. Even in the case that all clients are clean, the average accuracy decreases from 86.13 of FedAvg to 85.73 of Fed-FA and Fed-FA only has a performance decrease of about 0.40, which is much lower than other discarding aggregations. When there are malicious clients, Fed-FA has better clean performance than both FedAvg and other defenses. Besides, poisonous updates will also harm the learning, while Fed-FA will not harm the clean accuracy since it discards poisonous updates. To conclude, with our proposed Fed-FA, false positives have weak impacts on clean performance.

**Discarding a fixed number of clients can act as a strong defense.** Discarding aggregations are stronger baselines than non-discarding aggregations. In discarding aggregations, false positives have weak impacts on the clean performance, but the false negatives may poison the global model and fail the federated backdoor defense. Although discarding a fixed number of clients will cause false positives in malicious client detection, it can still act as a strong defense with little performance loss. Since the task of defending against federated language backdoors itself is difficult, we recommend discarding a fixed number of clients instead of distinguishing clean and malicious clients and only discarding poisonous updates. Our recommended method has a similar discarding protocol as both Fed-FA and other existing discarding aggregations [1, 27, 58].

**The proposed detection variants of discarding aggregations.** However, [6] argue that the influence of false positives is also crucial, especially for the cases when the backdoor defense is easier and the clean accuracy is crucial. Therefore, we also design detection variants for discarding aggregations for these cases when lower false positives are also important. In variants, we label the clients with distances or indicators (normalized to zero mean and one standard deviation) higher than a threshold as malicious clients. The threshold is tuned on the validation set for each task and defense. We validate the FAR (false acceptance rate), FRR (false rejection rate), P (precision), R (recall), F (F-1 score), ACC (accuracy), and MR (mean rank) on malicious client detection tasks of detection variants of discarding aggregations. The definitions of these indicators are:

$$\text{FAR} = \text{The probability when the benign client is regarded as a trojaned client,} \tag{71}$$

$$\text{FRR} = \text{The probability that the trojaned client is recognized as the benign client,} \tag{72}$$

$$\text{P} = \frac{\text{Number of true predicted malicious clients}}{\text{Number of total predicted malicious clients}}, \tag{73}$$

$$\text{R} = \frac{\text{Number of true predicted malicious clients}}{\text{Number of total true malicious clients}}, \tag{74}$$

$$\text{F} = \frac{2\text{PR}}{\text{P} + \text{R}}, \tag{75}$$

$$\text{ACC} = \text{The probability that a client is recognized correctly,} \tag{76}$$

$$\text{MR} = \text{Mean ranks of true malicious clients in the clients' distances or indicators.} \tag{77}$$

Table 11: Performance of detection variants of discarding aggregations. Variant denotes discarding according to the threshold, otherwise discarding $\lfloor \frac{n-1}{2} \rfloor$ clients. Lower MRs are better. Euclidean denotes Fed-FA with the Euclidean indicator. The best results are in **bold**.

| Method | FAR% | FRR% | P% | R% | F% | ACC% | MR |
|--------|------|------|-----|------|------|------|------|
| Random | 40.10 | 60.33 | 22.31 | 39.67 | 28.56 | 55.35 | 5.50 |
| Random (Variant) | 33.94 | 67.78 | 21.61 | 32.22 | 25.87 | 58.45 | 5.50 |
| M-Krum | 46.61 | 98.33 | 1.03 | 1.67 | 1.27 | 41.75 | 8.43 |
| M-Krum (Variant) | 51.13 | 97.78 | 1.25 | 2.22 | 1.60 | 38.38 | 8.43 |
| Euclidean | 51.29 | 97.22 | 1.55 | 2.78 | 1.99 | 38.38 | 8.49 |
| Euclidean (Variant) | 47.26 | 98.33 | 1.01 | 1.67 | 1.26 | 41.25 | 8.49 |
| Fed-FA | 32.26 | **33.33** | 37.50 | **66.67** | **48.00** | 67.50 | **3.78** |
| Fed-FA (Variant) | **10.16** | 70.00 | **46.15** | 30.00 | 36.36 | **76.38** | **3.78** |

Table 12: Influence of the synthetic dataset.

| Method | ACC | ASR |
|--------|-----|-----|
| FedAvg | 85.28±0.81 | 86.67±7.49 |
| Fed-FA with labeled dataset | 85.77±0.12 | 20.06±1.25 |
| **Fed-FA** | 85.70±0.18 | **19.51±1.90** |

Table 13: Robustness to distributed attacks.

| Attack | ACC | ASR |
|--------|-----|-----|
| FedAvg | 86.97 | 100.0 |
| Dim-Krum | 86.43 | 15.12 |
| Fed-FA | 86.57 | **13.45** |

**Detection performance of discarding aggregations and variants.** We report detection performance under multiple client numbers (0-4 of 10 clients) in Table 11. The detection variant of Fed-FA has lower false positives and higher false negatives, namely the detection variant tends to miss malicious updates and not to discard clean updates compared to Fed-FA. To conclude, there is a tradeoff between FAR and FRR and the detection variant of Fed-FA can be utilized in cases where false positive rates are crucial. Both Fed-FA and its variant have satisfying detection performance compared to random baseline and other discarding aggregations.

## D.2 The randomness of the synthetic dataset

As reported in Table 12, we randomly sample different texts as the labeled and synthetic datasets for different runs and the standard deviations of Fed-FA are low, which means that the randomness of the synthetic dataset does not influence the results much since Fed-FA only needs the Hessian scales instead of accurate Hessians.

## D.3 The robustness to distributed backdoor attacks

In addition to regularizers, we can adopt distributed backdoor attacks [44] to make malicious updates more stealthy as adaptive attacks. As shown in Table 13, we also validate that Fed-FA is robust to them.

