# OpenReview forum: "Fed-FA: Theoretically Modeling Client Data Divergence for Federated Language Backdoor Defense"
_NeurIPS.cc/2023/Conference — NeurIPS 2023 poster_

### Official Review · Reviewer_CgUg · 2023-07-05

**Soundness:** 4 excellent
**Presentation:** 3 good
**Contribution:** 3 good
**Rating:** 7
**Confidence:** 3

**Summary:**

The paper introduces a novel Federated F-Divergence-Based Aggregation (Fed-FA) algorithm to enhance defense against backdoor attacks in federated learning within NLP tasks. Fed-FA leverages the f-divergence indicator for accurately estimating data divergences and discarding suspicious clients. Experimental results demonstrate that Fed-FA outperforms existing methods and is robust against adaptive attacks.



**Strengths:**

* The paper addresses an important and timely issue - the robustness of federated learning systems against backdoor attacks.
* The proposed Fed-FA algorithm is novel and theoretically grounded, leading to effective detection and removal of suspicious clients.
* Fed-FA shows improved performance over existing methods and exhibits robustness against adaptive attacks, an important characteristic in a fast-evolving threat landscape.
* The paper provides a comprehensive evaluation, including ablation studies and tests under various conditions, further strengthening the validity of the results.

**Weaknesses:**

* The paper assumes that the data across clients is independent and identically distributed (IID), which may not be true in practical scenarios.
* The defense performance in non-IID cases was not as satisfactory as in IID cases.
* The paper lacks comparison with other state-of-the-art defense algorithms outside the parameter-distance-based category.

**Questions:**

* Could the authors provide further insight into how the f-divergence indicator performs compared to other divergence measures?
* Could the authors elaborate on potential real-world applications of Fed-FA?
* How does the proposed method perform compared to other state-of-the-art defenses, particularly those not based on parameter-distance?
* Can the authors comment on the scalability of the approach when the number of clients increases significantly?
* How does the method ensure privacy preservation while estimating data divergence across different clients?

**Limitations:**

* The approach heavily relies on accurate Hessian estimation. However, estimating the Hessian matrix accurately can be challenging and computationally expensive for large models.
* The authors do not discuss the potential privacy implications of their method, particularly when estimating data divergence across different clients.
* The computational overhead introduced by the proposed method is not discussed, leaving the efficiency of the proposed method in real-time applications unclear.
* There's a lack of analysis on the rate of false positives - honest clients that could be incorrectly identified as attackers.
* A primary limitation of Fed-FA and other existing methods is their reliance on the assumption of IID data across clients, which may not always hold in real-world scenarios.
* Another limitation is the lower defense performance of Fed-FA in non-IID settings compared to IID ones.
* The current method doesn't consider the semantics of parameter updates in the defense against backdoor attacks.

---

> ### Author Rebuttal · Authors · 2023-08-08
>
> We are grateful to the reviewer for positive comments. Here are our responses.
>
>
> [Q1] The paper assumes that the data across clients are independent and identically distributed (IID), which may not be true in practical scenarios.  The defense performance in non-IID cases was not as satisfactory as in IID cases.
>
> [A1] Non-IID cases are harder to defend than IID cases since the aggregation methods cannot distinguish malicious clients (with different distributions from clean clients) and clean clients with different distributions from others. As shown in our results, it is the limitation of our method and also the limitation of other defending algorithms. Besides, our Fed-FA still outperforms other algorithms in non-IID cases  as shown in Fig 3.
>
>
> [Q2] The paper lacks comparison with other state-of-the-art defense algorithms outside the parameter-distance-based category.
>
> [A2] We adopt both SOTA parameter-based [Dim-Krum] and non-parameter-based methods [RFA, Residual-based, Median, Foolsgold, CRFL] in experiments.
>
>
> [Q3] Could the authors provide further insight into how the f-divergence indicator performs compared to other divergence measures?
>
> [A3] As discussed in Appendix A. Other divergences are special cases of f-divergence. And in our algorithms (deduced in Theorem 1), the indicators are proportional to different divergences (depends on f''(1)）and do not influence the results.
>
>
> [Q4] Could the authors elaborate on potential real-world applications of Fed-FA?
>
> [A4] NLP federated backdoors are harder than CV backdoors to defend. Fed-FA can potentially enhance federated learning for NLP tasks. For example, if we want to train a spam emails classifier through the labels of spam reported by users without exposure the privacy of users, we can adopt federated learning to train this natural language model. However, some malicious clients will label some spam emails contraining their company names as non-spams, which acts as a backdoor attack and our proposed Fed-FA will detect these malicious clients and discard their updates.
>
>
> [Q5] Can the authors comment on the scalability of the approach when the number of clients increases significantly?
>
> [A5] When the number of clients grows, if the proportion of malicious clients remains the same (e.g. 1 malicious client out of 10 clients vs 10 malicious clients out of 100 clients), the defending performance will be similar.
>
>
> [Q6]  How does the method ensure privacy preservation while estimating data divergence across different clients? The approach heavily relies on accurate Hessian estimation. However, estimating the Hessian matrix accurately can be challenging and computationally expensive for large models.
>
> [A6] As reported in Table 4, Fed-FA performs similarly on labeled data and randomly labeled fake data. Therefore, Fed-FA only relies on the relative scales of Hessians and does not rely on the accurate estimation of Hessians. We use randomly labeled fake data to estimate Hessians instead of data from clients to avoid privacy exposure of clients.
>
>
> [Q7] The computational overhead introduced by the proposed method is not discussed, leaving the efficiency of the proposed method in real-time applications unclear.
>
> [A7] Note that the Hessian matrix is diagonal in our assumption. Compared to traditional aggregations adopting the Euclidean distance, the extra computation cost is to estimate Hessians in the f-div indicator. We mentioned in line 176-177: We synthetize 4 samples every class. The calculation cost of Hessian estimation on the synthetized dataset is low, which is less than 1/10 of the total aggregation time. Therefore the extra computation cost is relatively low.
>
>
> [Q8] There's a lack of analysis on the rate of false positives - honest clients that could be incorrectly identified as attackers.
>
> [A8]
> 1.   As mentioned in line 111-113 in related works and line 238-240 in main results, discarding methods are stronger baselines than non-discarding methods. Following other discarding baselines [Krum, Bulyan, Dim-Krum], we discard about 1/2 of clients, and the convergence is guaranteed in Theorem 3.
> 2.   Even in the case that all clients are clean, dropping 1/2 of clients will only cause a performance decrease of about 0.50 (average from FedAvg 86.3 to Fed-FA about 85.8 in our experiments, we will add it in revision).
> 3.   Besides, the backdoored updates will also harm the learning, removing about 1/2 of the clients will not harm the clean accuracy compared to Fed-Avg baselines and other defenses: As reported in Table 1 in our experiments, average clean accuracy: Fed-FA 85.70 vs Fed-Avg 85.28, other defenses 85.2-85.6).
> 4.   Therefore, dropping 1/2 of clients is necessary for a strong defense and may not harm the clean performance much in NLP backdoor defense since the defending task itself is hard.
>
>
> (References are in global response)

---

> > ### Comment · Reviewer_CgUg · 2023-08-14
> >
> > Thanks to authors for their thorough response. My concerns have been answered and I am willing to upgrade my score to "accept".

---

> > > ### Author Response · Authors · 2023-08-15
> > >
> > > Thanks. We will improve our revision according to the reviewers' concerns.

---

### Official Review · Reviewer_z5C7 · 2023-07-06

**Soundness:** 3 good
**Presentation:** 3 good
**Contribution:** 3 good
**Rating:** 5
**Confidence:** 4

**Summary:**

The paper proposes a new algorithm called Federated F-Divergence-Based Aggregation (Fed-FA) to defend against backdoor attacks in natural language processing (NLP) tasks. Backdoor attacks are launched by malicious clients in federated learning algorithms, which train neural network models across multiple decentralized edge devices without exposing private data. Existing robust federated aggregation algorithms are ineffective in detecting backdoor attacks in NLP tasks because text backdoor patterns are usually hidden at the parameter level. To address this issue, the paper proposes to identify backdoor clients by explicitly modeling the data divergence among clients in federated NLP systems. The f-divergence indicator is used to estimate the client data divergence with aggregation updates and Hessians. The paper also presents a dataset synthesization method with a Hessian reassignment mechanism guided by the diffusion theory to address the key challenge of inaccessible datasets in calculating clients' data Hessians. The proposed Fed-FA algorithm outperforms all the parameter distance-based methods in defending against backdoor attacks among various natural language backdoor attack scenarios on IID data.

**Strengths:**

1.This paper is easy to follow. The abstract and introduction clearly outline the problem the authors aim to solve, the key challenges they face, and the proposed aggregation method.

2.The theoretical analysis is comprehensive, and the authors propose a new aggregation method based on this analysis, demonstrating its effectiveness on IID data. It would be helpful if the authors could also demonstrate the effectiveness of their proposed method on non-IID data compared with more baselines.

**Weaknesses:**

1.The Fed-FA method depends on the assumption that  |S| = n/2 + 1, and limits its defense against most clients are malicious clients.

2.Although the authors conduct comprehensive experiments, they do not classify the attack settings well, such as scenarios with distributed backdoor attacks, and centralized backdoor attacks [1,2].



[1] DBA: Distributed Backdoor Attacks against Federated Learning

[2] Bkd-FedGNN: A Benchmark for Classification Backdoor Attacks on Federated Graph Neural Network

**Questions:**

1.Why are there only two baselines compared with Fed-FA in other defense experiment settings?

**Limitations:**

1.Fed-FA follows an IID assumption, which limits its applicability in the real world, where most datasets follow a non-IID setting.

2.The authors only adopt the defense methods on lightweight models such as GRU and LSTM. However, these models may still have original vulnerabilities. Large language models are not included in their analysis.

---

> ### Author Rebuttal · Authors · 2023-08-08
>
> We appreciate the reviewer’s detailed comments. Here are our responses.
>
>
> [Q1]The Fed-FA method depends on the assumption that |S| = n/2 + 1, and limits its defense against most clients are malicious clients.
>
> [A1] Theoretically, if most clients are malicious, the server can not defend against the attack since in the view of the server, malicious clients are normal (>1/2) and clean clients are abnormal (<1/2). In most practical scenarios, the proportion of clients with backdoors is relatively low, which is also the assumption of existing classic works such as [Krum], [Bulyan], [Dim-Krum].
>
>
> [Q2] Although the authors conduct comprehensive experiments, they do not classify the attack settings well, such as scenarios with distributed backdoor attacks, and centralized backdoor attacks.
>
> [A2] Since our paper focuses on NLP tasks, we follow the classification of NLP backdoor defense [Dim-Krum] and adopts four types of NLP backdoors: EP, badword, badsent, hidden killer. We think the attack settings can cover typically NLP backdoor attacks. Distributed backdoor attacks and centralized backdoor attacks are another classification method of backdoors. Most of our attacks can be treated as centralized attacks when malicious client=1. Besides, when client>1, EP attacks on different clients will choose different trigger words that can both be malicious trigger words and can be seen as a distributed backdoor attack. Fed-FA can also defend it. We will add these discussions in the revision.
>
>
> [Q3] Large language models are not included in their analysis.
>
> [A3] Simulating the federated learning process of a large-scale model on multiple clients is quite computationally expensive. Classic federated defending algorithms are also evaluated on small-scale models. For example, [Krum] adopt the CNN model; [CRFL] adopt multi-class logistic regression; [Residual-based] adopts a two-layer convolutional neural network; [RFA] adopt a linear model and a convolutional neural network; [Dim-Krum] adopt GRU, LSTM and CNN models. Moreover, our proposed Fed-FA is model-agnostic and just filters harmful gradients involved in aggregation, so it can be extended to large language models.
>
>
> [Q4] IID assumption may be not true in the real world.
>
> [A4] Non-IID cases are harder to defend than IID cases since the aggregation methods cannot distinguish malicious clients (with different distributions from clean clients) and clean clients with different distributions from others. As shown in our results, it is the limitation of our method and also the limitation of other defending algorithms. Besides, our Fed-FA still outperforms other algorithms in non-IID cases as shown in Fig 3.
>
>
> (References are in global response)

---

> > ### Comment · Reviewer_z5C7 · 2023-08-18
> > **Response to Authors**
> >
> > Thank you for the authors' response. I appreciate their efforts in addressing most of my concerns. While I have no further questions, I would like to reiterate my viewpoint that conducting the experiments under the LLM settings with FL would enhance the study's relevance, particularly considering the recent developments in related works. The authors' response is indeed valuable, and I believe incorporating experiments in the LLM context could provide additional insights and strengthen the overall contribution of the research.
> >
> > 1.  Towards Building the Federated GPT: Federated Instruction Tuning
> >  2. FedPETuning: When Federated Learning Meets the Parameter-Efficient Tuning Methods of Pre-trained Language Models

---

> > > ### Author Response · Authors · 2023-08-19
> > >
> > > Thanks for your helpful comments. Though classic federated defending algorithms are also evaluated on small-scale MLP, CNN or RNN models [Krum, CRFL, Residualbased, RFA, DimKrum] due to computation cost limit, we agree that evaluating on Transformers or large language models will benifit the work. Our proposed Fed-FA is model agnostic and just filters harmful gradients involved in aggregation, thus it can be extended to large language models. We will add the experiments of transformers and distilled language models in revision.

---

### Official Review · Reviewer_2F5q · 2023-07-06

**Soundness:** 3 good
**Presentation:** 2 fair
**Contribution:** 3 good
**Rating:** 5
**Confidence:** 4

**Summary:**

This paper proposes a defense method against backdoor attacks in federated learning for NLP tasks. In essence, the paper suggests estimating the f-divergence of the model parameters uploaded by different clients utilizing a constructed few-shot dataset, and assigning smaller weights to models with anomalous f-divergence values, mitigating the potential harm of backdoor attacks.

**Strengths:**

1. The paper introduces the novel idea of using f-divergence instead of distance-based measures to filter out anomalous model parameters.
2. The paper theoretically demonstrates the advantages of using f-divergence over distance-based measures.

**Weaknesses:**

1. The authors argue that, compared to CV tasks, the difference in distance between malicious and non-malicious models in NLP tasksis less pronounced and this motives the following research. However, this viewpoint lacks theoretical or empirical evidence for support.
2. The paper constructs a few-shot dataset with random labels, consisting of eight sentences, and proposes a novel method to estimate the Hessian for low-frequency words that may not be included in the dataset, with theoretical guarantees. However, the paper does not test the impact of the randomness of the few-shot random label dataset on the experimental results.
3. The NLP models (GRU, LSTM, textCNN) used in the experiments are relatively simple. Is it possible to test on larger models, such as Bert, RoBerta, and GPT2?
4. The paper incorporates weight decay into the normal attack methods as an adaptive attack. Is this assumption too weak? Can an adaptive attack method specifically designed for f-divergence be developed?

**Questions:**

see above

**Limitations:**

see above

---

> ### Author Rebuttal · Authors · 2023-08-08
>
> We appreciate the reviewer’s detailed comments. Here are our responses.
>
>
> [Q1] The authors argue that, compared to CV tasks, the difference in distance between malicious and non-malicious models in NLP tasks is less pronounced and this motivates the following research. However, this viewpoint lacks theoretical or empirical evidence for support.
>
> [A1]  Existing baselines [Krum, CRFL, Residual-based] can form a satisfying defense in CV tasks but perform worse in NLP tasks (in both [Dim-Krum] and our experiments). [Dim-Krum] revealed that NLP federated backdoors are hard to defend than CV, and analytical experiments in [Dim-Krum] revealed that the reason may lie in that NLP backdoor updates are more stealthy than CV (namely poisonous updates are closer to clean updates in NLP tasks). In our paper, we also pointed out the reason may be that NLP backdoors can be very local and stealthy on only a few parameters or features (e.g. embeddings of trigger word), which will not cause siginificant statistical changes in parameter distances and thus NLP backdoors are harder than CV backdoors.
>
>
> [Q2] The paper constructs a few-shot dataset with random labels, consisting of eight sentences, and proposes a novel method to estimate the Hessian for low-frequency words that may not be included in the dataset, with theoretical guarantees. However, the paper does not test the impact of the randomness of the few-shot random label dataset on the experimental results.
>
> [A2] We randomly choose different few-shot sentences from Wikipedia in different runs. The randomness does not influence the results much since Fed-FA only needs the Hessian scales instead of accurate Hessian estimations. As reported in Table 4, Fed-FA performs similarly on labeled data and randomly labeled fake data.  We will add these discussions in the revision.
>
>
> [Q3] The NLP models (GRU, LSTM, textCNN) used in the experiments are relatively simple. Is it possible to test on larger models, such as Bert, RoBerta, and GPT2?
>
> [A3] Simulating the federated learning process of a large-scale model on multiple clients is quite computationally expensive. Classic federated defending algorithms are also evaluated on small-scale models. For example, [Krum] adopt the CNN model; [CRFL] adopt multi-class logistic regression; [Residual-based] adopts a two-layer convolutional neural network; [RFA] adopt a linear model and a convolutional neural network; [Dim-Krum] adopt GRU, LSTM and CNN models. Moreover, our proposed Fed-FA is model-agnostic and just filters harmful gradients involved in aggregation, so it can be extended to other models.
>
>
> [Q4] The paper incorporates weight decay into the normal attack methods as an adaptive attack. Is this assumption too weak? Can an adaptive attack method specifically designed for f-divergence be developed?
>
> [A4] The weight decay on |poisoned weight-initial weight|^2 can be an adaptive attack because Krum algorithms [Krum], including Fed-FA, detect poisonous clients by parameter distance and attacks with smaller partameter distances are more stealthy. The weight decay as an adaptive attacks is proposed in [Dim-Krum]. We also try another version of adaptive attacks, where the decaying loss has the weights of Hessian, namely $\sum_k Hessian_k*(poisoned weight_k-initial weight_k)^2$. Fed-FA can also defend it and the results are similar to weight decay. We will discuss it in revision.
>
>
> (References are in global response)

---

> > ### Author Response · Authors · 2023-08-21
> >
> > Dear Reviewer,
> >
> > Thanks for your reply.
> >
> > 1.More details to Q2: With randomly chosen labled few-shot dataset, average ACC ± STD and ASR ± STD of Fed-FA on different runs are 85.77 ±  0.12 and 20.06 ± 1.25. On randomly chosen unlabeled few-shot dataset, average ACC ± STD and ASR ± STD of Fed-FA on different runs are 85.70 ±  0.18 and 19.51 ± 1.90 (10 runs, few-show dataset = 8 examples, average ACC are also reported in Table 4 in the paper). There is no statistically significant difference in the experimental effect of estimating Hessian using labeled data and unlabeled data. Therefore, the randomness or inaccurate estimation of Hessian will not affect the defense performance since we only need rough magnitudes of Hessians as weights in F-div estimation.
> >
> > 2.More details to Q3: Thanks for your advise. We tried a more deeper 3-layer bidirectional LSTM. As shown in below:
> >
> > |Defense | 3-layer LSTM |Avg ACC  |Avg ASR | 1-layer LSTM |Avg ACC  |Avg ASR |
> > | :-----| :----: | :----: | :----: | :----: | :----: | :----: |
> > | Fed-Avg |3-layer LSTM | 84.21| 89.83 |1-layer LSTM | 83.49| 90.51 |
> > | Dim-Krum | 3-layer LSTM |83.01 | 35.22 |1-layer LSTM |82.91 | 33.08 |
> > | Fed-FA | 3-layer LSTM |84.91 | **22.08** | 1-layer LSTM |84.39 | **22.11** |
> >
> > The trands of results of 3-layer bidirectional LSTMs are consistent to single-layer models reported in paper. (Besides, STDs of ACC are about 0.1%-0.2% and STDs of ASR are about 1%-2%; The STD trends are consistent to STD trends reported in the Appendix of our original paper). Therefore, our proposed Fed-FA is model-agnostic and just filters harmful gradients involved in aggregation, and it can be extended to larger models. We will add the experiments of larger models such as transformers and language models in revision.

---

> > > ### Comment · Reviewer_2F5q · 2023-08-21
> > >
> > > I thank the authors for the response and additional experiments. Though I still think experiments on larger models are necessary, I think the authors have addressed most of my concerns and thus I will increase my score to 5.

---

> > > > ### Author Response · Authors · 2023-08-21
> > > >
> > > > Thanks for your valuable comments. Fed-FA can be extended to larger models and we will add these experiments in revision.

---

### Official Review · Reviewer_hfUd · 2023-07-20

**Soundness:** 3 good
**Presentation:** 2 fair
**Contribution:** 3 good
**Rating:** 5
**Confidence:** 3

**Summary:**

The paper proposes a new defense method against backdoor attacks in federated learning for NLP tasks. Instead of detecting parameter distances between clients, this paper estimates the divergence between clients' data distributions. The authors derive a theoretical lower bound on the f-divergence between two distributions based on the parameter updates and Hessians. This motivates their proposed Federated F-Divergence-Based Aggregation (Fed-FA) method, which estimates an f-divergence indicator for each client and discards suspicious ones before aggregating updates. Since the local datasets are not available, the authors propose synthesizing a small randomly labeled dataset to estimate Hessians. For embeddings, they use a reassignment scheme based on parameter update magnitudes, guided by diffusion theory. Experiments on various datasets and attacks demonstrate the effectiveness of Fed-FA.

**Strengths:**

1. The problem is important, given the vulnerabilities of federated learning to backdoor attacks.
2. The method is theoretically grounded. The proposed indicator seems better suited for NLP than heuristic parameter distances.
3. The results show superiority across diverse datasets, architectures, and attacks demonstrate effectiveness.

**Weaknesses:**

1. The paper can be enhanced by improving the clarity of the writing.
- Despite providing a brief background on federated learning paradigm, the unique challenges of defending against backdoor attacks in federated learning are not clearly demonstrated. Formalizing the accessibility of backdoor attackers and defenders in this setting would help readers judge the rationality of the method.
- Some important details require further explanation. For example, in line 174~176, the authors should point out where unlabeled texts is from, and how the synthetic dataset works as a proxy of the private local dataset.
2. The evaluation setup is not rigorous.
- All experiments in this paper assume the existence of poisoned clients, but this may not match real-world scenarios where it is unknown if any client in the federated system has been injected backdoors. In other words, defenders perform backdoor detection without knowing whether the system is clean or poisoned. Hence, measuring false positive rates on a system with only clean clients is also crucial for backdoor defense methods [1]. A practical defense method should not discard many clean clients. Otherwise, the performance of the system may suffer degradation.

I would like to raise the score if the authors can address my concerns.

References:
[1] A Unified Evaluation of Textual Backdoor Learning: Frameworks and Benchmarks. Cui et al.

**Questions:**

Please refer to Weaknesses.

**Limitations:**

The authors have pointed out the limitation in non-IID cases. Though the problem is not well solved, the proposed method is still much better than its baseline methods.

---

> ### Author Rebuttal · Authors · 2023-08-08
>
> We appreciate the reviewer’s helpful comments. Here are our responses.
>
>
> [Q1] Since it is unknown if any client in the federated system has been injected backdoors, measuring false positive rates on a system with only clean clients is also crucial for backdoor defense methods [1]. A practical defense method should not discard many clean clients. Otherwise, the performance of the system may suffer degradation.
>
> [1] A Unified Evaluation of Textual Backdoor Learning: Frameworks and Benchmarks. Cui et al.
>
> [A1]
> 1.Thanks for your advise, we will compare Fed-FA with other baselines in the settings in [1] in the revision.
>
> 2.The reasons for discarding 1/2 of suspicious clients in our paper lie that:
>
> a.As mentioned in line 111-113 in related works and line 238-240 in main results, discarding methods are stronger baselines than non-discarding methods. Following other discarding baselines [Krum, Bulyan, Dim-Krum], we discard about 1/2 of clients, and the convergence is guaranteed in Theorem 3.
>
> b.Even in the case that all clients are clean, dropping 1/2 of clients will only cause a performance decrease of about 0.50 (average from FedAvg 86.3 to Fed-FA about 85.8 in our experiments, we will add it in revision).
>
> c.Besides, the backdoored updates will also harm the learning, removing about 1/2 of the clients will not harm the clean accuracy compared to Fed-Avg baselines and other defenses: As reported in Table 1 in our experiments, average clean accuracy: Fed-FA 85.70 vs Fed-Avg 85.28, other defenses 85.2-85.6).
>
> d.Therefore, dropping 1/2 of clients is necessary for a strong defense and may not harm the clean performance much in NLP backdoor defense since the defending task itself is hard.
>
>
> [Q2] The paper can be enhanced by improving the clarity of the writing.
>
> [A2] Thanks for your advise, we will improve our writing and clarify these important details mentioned in the revision.
>
>
> (References are in global response)

---

> > ### Comment · Reviewer_hfUd · 2023-08-16
> > **Response should be more informative**
> >
> > Dear authors,
> >
> > Thanks for your reply. I'm willing to raise my score if you can address my concerns. However, your answer is too vague, especially your answer to Q2. Can you explain your additional experiments more clearly and how you will improve the writing?

---

> > > ### Author Response · Authors · 2023-08-17
> > >
> > > Dear Reviewer,
> > >
> > > Thanks for your reply.
> > >
> > > Since as [1] stated, false positive rates are crucial, we plan to add experiments to verify the effects of the false positive in backdoor detection as rebuttal A1.2.b: Namely, even Fed-FA drops 1/2 clean clients when all clients are clean because the clients discarded in different rounds are different, the parameters of each client still have a chance to be learned by the server. Compared with other defense algorithms or FedAvg, the performance loss is limited. Some extra experiments are already conducted and results are reported in rebuttal A1.2.b: "even if all clients are clean, dropping 1/2 of clients will only cause a performance decrease of about 0.50 (average ACC on LSTM/GRU/TextCNN from FedAvg 86.3 to Fed-FA about 85.8 in our experiments, and other defenses also report similar or even larger ACC decreases)". We will add these discussions to revision.
> > >
> > > Besides, as mentioned in rebuttal A1.1, [1] proposed that the false acceptance rate (FAR) that misclassifies poisoned samples as normal and the false rejection rate (FRR) that misclassifies normal samples as poisoned are also crucial. Similarly, we will adopt a detection variant that aims to detect poisoned clients using a threshold method and labels clients with higher F-div than the threshold (determined on the dev set) as the poisoned clients (rather than 1/2 of clients) and calculate the FRR and FAR of Fed-FA compared to detection variants of other defenses.
> > >
> > > For writing, thanks to the helpful comments of the reviewers, we have realized that we can improve the presentation by adding or extending these discussions: (1) the unique challenge of NLP backdoors: existing baselines [Krum, CRFL, Residual-based] can form a satisfying defense in CV tasks but perform worse in NLP tasks (in both [Dim-Krum] and our experiments). [Dim-Krum] revealed that NLP federated backdoors are hard to defend than CV, and analytical experiments in [Dim-Krum] revealed that the reason may lie in that NLP backdoor updates are more stealthy than CV, namely poisonous updates are closer to clean updates in NLP tasks. In our paper, we also pointed out the reason may be that NLP backdoors can be very local and stealthy on only a few parameters or features (e.g. embeddings of trigger word), which will not cause significant statistical changes in parameter distances and thus NLP backdoors are harder than CV backdoors. (2) The Hessian estimation cost is low, only 1/10 time of total aggregation time on the server (do not include training time on the server); (3) The source of the text (sampled from Wikipedia) and the influence of randomness and inaccurate estimation of Hessian caused by random text and privacy data deviation (the randomness or inaccurate estimation of Hessian will not affect the defense performance since we only need rough magnitudes of Hessians as weights in F-div estimation).

---

> > > > ### Comment · Reviewer_hfUd · 2023-08-21
> > > > **Response to authors**
> > > >
> > > > Dear Authors,
> > > >
> > > > Thanks for your reply. I would like to raise my score to 5.

---

> > > > > ### Author Response · Authors · 2023-08-21
> > > > >
> > > > > Thanks. We will improve our revision according to the reviewers' valuable suggestions.

---

> ### Comment · Area_Chair_Cy4d · 2023-08-21
> **Check the response from the authors**
>
> Dear reviewer,
>
> The authors have reacted regarding to your concerns about extra experiments and writing. Could you please check the response and respond to them?
>
> Thanks
>
> AC

---

### Official Review · Reviewer_hSLg · 2023-07-27

**Soundness:** 3 good
**Presentation:** 2 fair
**Contribution:** 3 good
**Rating:** 5
**Confidence:** 2

**Summary:**


This paper studies how to identify the backdoor in Federated Learning (FL) which is achieved by 'explicitly modeling the data divergence among clients'. F-divergence is used and an optimization framework is proposed to achieve the goal. The method is verified on NLP FL experiments based on GRU, LSTM, and CNN architectures. Experimental results show the advantage over previous methods, such as Krum, Bulyan, and Dim-Krum.

**Strengths:**

+ The paper proposes to use F-divergence to extend the search space for distance measurement, which potentially assists in identifying the backdoored clients.

+ A method is proposed to select reliable clients, based on the infimum of the divergences.

+ The experiments show that Fed-FA works better than a series of Krum-based baselines.



**Weaknesses:**

+ Because Eucleadian distance is a special case of f-divergence, the success of f-divergence is quite predictable.

+ The writing of the paper needs improvement.

+ The increased computational cost is not discussed. Will this method work on large-scale models?

+ The method seems always remove more than 1/2 of the contributions by clients, which may lead to slower convergence.

**Questions:**


How do you decide $\sigma$ in Eqn 2?

+ Minor
  + L2: 'without private data exposure' may not be true.
  + L30: NLP tasks 'are harder to defend against than vision tasks;' may not be true.
  + 'Infimum' -> 'Inf' will be enough.
  + The author should lead readers to Appendix when it is necessary to check the details.
  + 'f' (client number) in Theorem 3 is mixed with other functions f.
  + Fed-FA should be clearly split with $I_{F-Div}$ in algorithm.
  + It would be appreciated to link the operations in Algorithm 1 to the theorems.

**Limitations:**



The defense is assumed on FedAvg. How will this method work on other gradient aggregation methods?

The method is verified on GRU, LSTM, CNN + NLP tasks? What will be its potential on Transformer and/or CV tasks? I did not foresee any reason it will fail on continuous spaces.

---

> ### Author Rebuttal · Authors · 2023-08-08
>
> We appreciate the reviewer’s detailed comments. Here are our responses.
>
> [Q1] Because Eucleadian distance is a special case of f-divergence, the success of f-divergence is quite predictable.
>
> [A1] The success of the Euclidean distance lacks theoretical evidence, and our theoretical analysis addresses this issue. Our work deepens the understanding of this problem and extends the paradigm of optional distributions: distributions that conform to f-divergence can reasonably model data differences, while the Euclidean distance is just one particular case of f-divergence.
>
>
> [Q2] The increased computational cost is not discussed.
>
> [A2] Compared to traditional aggregations adopting the Euclidean distance, the extra computation cost is to estimate Hessians in the f-div indicator. We mentioned in line 176-177: We synthetize 4 samples every class. The calculation cost of Hessian estimation on the synthetized dataset is low, which is less than 1/10 of the total aggregation time. Therefore the extra computation cost is relatively low.
>
>
> [Q3] Will this method work on large-scale models?
>
> [A3] Our proposed Fed-FA is model-agnostic and there is no evidence that our method fails on large models. However, we do not test our method on large-scale models such as Transformer or BERT since simulating the federated learning process of a large-scale model on multiple clients is quite computationally expensive. Classic federated defending algorithms are also evaluated on small-scale models. For example, [Krum] adopt the CNN model; [CRFL] adopt multi-class logistic regression; [Residual-based] adopts a two-layer convolutional neural network; [RFA] adopt a linear model and a convolutional neural network; [Dim-Krum] adopt GRU, LSTM and CNN models.
>
>
> [Q4] The method seems always remove more than 1/2 of the contributions by clients, which may lead to slower convergence.
>
> [A4] As mentioned in line 111-113 in related works and line 238-240 in main results, discarding methods are stronger baselines than non-discarding methods. Following other discarding baselines [Krum, Bulyan, Dim-Krum], we discard about 1/2 of clients. The convergence may be slower (which is also the cost of other discarding baselines [Krum, Bulyan, Dim-Krum]), but is guaranteed in Theorem 3. Besides, the backdoored updates will also harm the learning, removing about 1/2 of the clients will not harm the clean accuracy compared to Fed-Avg baselines and other defenses: As reported in Table 1 in our experiments, average clean accuracy: Fed-FA 85.70 vs Fed-Avg 85.28, other defenses 85.2-85.6).
>
>
> [Q5] How do you decide $\sigma$ in Eqn 2?
>
> [A5]  Eq 2 does not have $\sigma$. Do you mean $\delta$? As mentioned in line 151-154, when q denotes the distribution on the client k, $\delta=\theta^{(k)} − \theta^{Avg}$.
>
>
> [Q6] The defense is assumed on FedAvg. How will this method work on other gradient aggregation methods?
>
> [A6] The core of our method is to prevent gradient participation from clients with backdoors during aggregation. This is equivalent to optimizing the input of the aggregation algorithm, which is independent of the aggregation algorithm itself and can be applied to general aggregation algorithms. In addition, FedAvg is a very popular and general federated learning aggregation method, which is commonly used as a benchmark method in existing works [Krum, Residual-based, Dim-Krum].
>
>
> [Q7] The method is verified on GRU, LSTM, CNN + NLP tasks? What will be its potential on Transformer and/or CV tasks? I did not foresee any reason it will fail on continuous spaces.
>
> [A7] We focus on NLP since NLP tasks are harder than CV tasks [Dim-Krum], since existing baselines [Krum, CRFL, Residual-based] can form a satisfying defense in CV tasks but perform worse in NLP tasks (in both [Dim-Krum] and our experiments). We think the proposed Fed-FA can also form a strong defense on CV tasks since they are easier than NLP tasks. We will add extra experiments in the revision. Moreover, our proposed Fed-FA is model-agnostic and just filters harmful gradients involved in aggregation, so it can be extended to large-scale models like Transformer, BERT or GPT.  The reason we chose to conduct experiments on LSTM/GRU/CNN models is that these smaller models are more suitable for deployment on client-side with limited computing power, which is also a mainstream practice in existing works [CRFL, Residual-based, Dim-Krum].
>
>
> (References are in global response)

---

> > ### Comment · Reviewer_hSLg · 2023-08-16
> > **Thank you for your responses**
> >
> > > To: Q3: Will this method work on large-scale models?
> >
> > I still think it is worth evaluating your methods on Transformer if you focus on NLP tasks. Diffusion Model is of great interest if you are working on CV tasks.
> >
> > > To: "NLP tasks are harder than CV tasks"
> >
> > I cannot agree with this argument. Both areas have their own research challenges.

---

> > > ### Author Response · Authors · 2023-08-17
> > >
> > > Dear Reviewer,
> > >
> > > Thanks for your reply.
> > >
> > > 1. To Q3: Transformers are indeed very important in NLP. However, due to limitations in computational resources on the client side, the transformer model is not the primary experimental model in existing NLP federated learning [A7]. Furthermore, our method theoretically models data divergence to detect anomalous clients, which is independent of the experimental model. Therefore, there are no theoretical risk to transferring this method to other models like transformers. We will add the experiments of transformers and distilled language models in revision.
> > >
> > > 2. To "NLP tasks are harder than CV tasks". Sorry for the ambiguous abbreviation in [A7], "NLP tasks are harder than CV". Here we mean that for existing parameter-distance-based defense algorithms, defending federated NLP backdoors is harder than CV backdoors, which is verified in experiments and explained both [A7] and our paper. Thus we focus on NLP defense.

---

### Author Rebuttal · Authors · 2023-08-08

We sincerely thank the reviewers for their helpful comments. We respond to the concerns of reviewers respectively. Here are the References and they are named [algorithm name]:

Reference

[Krum]   Blanchard, P., Mhamdi, E.M.E., Guerraoui, R., Stainer, J.: Machine learning with adversaries: Byzantine tolerant gradient descent. In NeurIPS 2017.

[Bulyan] Mhamdi, E.M.E., Guerraoui, R., Rouault, S.: The hidden vulnerability of distributed learning in byzantium. In ICML 2018.

[Dim-Krum] Zhang, Z., Su, Q., Sun, X.: Dim-krum: Backdoor-resistant federated learning for NLP with dimension-wise krum-based aggregation. In Findings of EMNLP 2022.

[Median] Chen, X., Chen, T., Sun, H., Wu, Z.S., Hong, M.: Distributed training with heterogeneous data: Bridgin gmedian- and mean-based algorithms. In NeurIPS 2020.

[Foolsgold] Fung, C., Yoon, C.J.M., Beschastnikh, I.: The limitations of federated learning in sybil settings. In RAID 2020.

[RFA] Pillutla, V.K., Kakade, S.M., Harchaoui, Z.: Robust aggregation for federated learning. Arxiv:1912.13445.

[CRFL] Xie, C., Chen, M., Chen, P., Li, B.: CRFL: certifiably robust federated learning against backdoor attacks. In ICML 2021.

[Residual-based] Fu, S., Xie, C., Li, B., Chen, Q.: Attack-resistant federated learning with residual-based reweighting. Arxiv:1912.11464.

[FedAvg] McMahan, B., Moore, E., Ramage, D., Hampson, S., y Arcas, B.A.: Communication-efficient learning of deep networks from decentralized data. In AISTATS 2017.

---

### Author Response · Authors · 2023-08-15

Dear PC and AC members of NeurIPS23,

We would like to express our sincere gratitude for your efforts in organizing the conference. We are writing this letter to draw your attention to the review comments provided by Reviewer 2F5q and Reviewer hfUd. It seems that they may have misunderstood certain aspects of our work, which we have clarified in our rebuttal. However, we have not received any response thus far, and we are eager to engage in a constructive dialogue with them to address any misunderstandings. We kindly request your assistance in facilitating this communication.

Thank you for your attention.

Sincerely, Authors of Submission2030.

---

> ### Comment · Area_Chair_Cy4d · 2023-08-21
> **Reviewers have responded to your rebuttals**
>
> Dear authors,
>
> Thanks for the reminding message. In fact, reviewers have had a discussion regarding to your rebuttals.
>
> Here is the response from 2F5q:
> "I thank the authors for the detailed response. While my concerns are partially resolved, I still feel the experiments part is a bit weak and want to see more concrete results.
>
> For Q2, can you show the results on avg ACC and ASR along with the standard deviations when using different randomly chosen few-shot sentences?
>
> For Q3, I understand that large models take time to compute especially in federated settings, however, the current experiments on GRU/LSTM are based on single-layer models which is too small for practical use. I am hoping the authors could provide more concrete results on larger and popular models (like attention-based models) to make it more convincing.
>
> I am willing to raise my score if those concerns can be resolved."
>
> You cannot read his/her comments, probably due to he/she did not tip "authors" when submitting the response,
>
> Regarding to hfUd, I see you have a discussion about the experiment setup. The major concern lies in the extra experiments. I believe more experiments could further improve the work, but for fairness, please understand reviewers should decide "accept/reject" mainly based one the current form of the paper.  I will also urge the reviewer to respond to your new comments.
>
> Thanks
>
> AC

---

> > ### Author Response · Authors · 2023-08-21
> >
> > Dear AC,
> >
> > Thanks for your notification about the response from 2F5q. We have reponded the Reviewer 2F5q now.
> >
> > Thanks to all Reviewers, AC, SAC for their contributions to conference review again.
> >
> > Sincerely, Authors of Submission2030.

---

### Decision · Program_Chairs · 2023-09-21

**Decision:**

Accept (poster)

**Comment:**

The paper studies how to defend backdoor attacks in federated learning in NLP tasks. The authors theoretically derive the f-divergence indicator, and present a Federated F-Divergence-Based Aggregation (Fed-FA) algorithm that leverages the f-divergence indicator to detect and discard suspicious clients. Empirical studies with various models across different datasets indicate the effectiveness of the proposed method.

Some reviewers raised concerns regarding to the assumptions of the method, such as the number of clean clients and the way that the synthesized data is constructed, and the authors clarified with some extra results. Finally, all reviewers feel positive to the work. My comment is that this is an interesting work with minor issues that can be solved in the revision. Therefore, I recommend an "acceptance" and believe that the authors will involve all improvement they promise in their rebuttals in the final version.